# Differential splice isoforms of mouse CDK2 play functionally redundant roles during mitotic and meiotic division

Nathan Palmer[1], Nisan Ece Kalem-Yapar[2], Hanna Hultén[2], Umur Keles[2], S. Zakiah A. Talib[3], Jin Rong Ow[4], Tommaso Tabaglio[4], Christine M. F. Goh[4], Li Na Zhao[2], Ernesto Guccione[5], Kui Liu[6,7] and Philipp Kaldis[2,*]

## ABSTRACT

In most mammals, the cell cycle kinase cyclin-dependent kinase 2 (CDK2) is expressed as two major isoforms as a result of alternative splicing. The shorter CDK2 isoform, CDK2S, is expressed constitutively during the cell cycle and can be detected in several tissues. In contrast, the longer isoform, CDK2L, shows preferential expression in meiotically dividing cells and upon S-phase entry in the mitotic cycle. Both CDK2L and CDK2S form heteromeric complexes with cyclins A2 and E1 *in vitro*. However, complexes composed of each isoform differ considerably in their kinase activity towards known CDK substrates. It is currently unknown whether the long and short isoforms of CDK2 play functionally different roles *in vivo* during either mitotic or meiotic divisions as conventional knockout methodology deletes both of the isoforms. Therefore, we generated mice expressing only CDK2S or CDK2L and found that both CDK2L and CDK2S are sufficient to support both mitotic and meiotic division when expressed in the absence of the other. These data contribute to the explanation of the apparent tolerance of the evolutionary loss of CDK2L expression in humans.

KEY WORDS: CDK2, Mitosis, Meiosis, Gene function, Splicing, Mouse

## INTRODUCTION

Alternative splicing enables multiple protein isoforms to be generated from a single transcriptional unit, thus conferring added complexity to the human transcriptome and proteome (Consortium et al., 2007). Although the majority of predicted alternative transcripts are thought not to be translated into functional proteins, in many cases proteins arising from alternative spliced isoforms have been shown to play crucial and often divergent roles in important cellular processes (Yang et al., 2016). In this study, we investigate the function of an alternatively spliced isoform of cyclin-dependent kinase 2 (CDK2).

During mammalian cell division, cyclin-dependent kinase 2 (CDK2) phosphorylates key proteins to promote G1/S transition and DNA replication during S phase. These actions of CDK2 are executed in association with the E- and A-type cyclins, respectively (Fang and Newport, 1991; Koff et al., 1992). Despite such central cell cycle-oriented functions, CDK2 is not essential for cellular division in several mammalian cell lines (Tetsu and McCormick, 2003), and $Cdk2^{-/-}$ mice are viable (Berthet et al., 2003; Ortega et al., 2003). The lack of proliferative defects in $Cdk2^{-/-}$ cells is due to functional redundancy between CDK2 and the closely related CDK1, which is essential for cellular division (Diril et al., 2012; Santamaria et al., 2007; Satyanarayana et al., 2008), as well as CDK4 (Berthet et al., 2006). In the absence of CDK2, CDK1 can fulfil the roles of CDK2 by binding to E-, A- and D-type cyclins (Aleem et al., 2005). Unlike during mitotic division, CDK2 is essential for meiotic divisions. In meiotically dividing cells, CDK1 is unable to compensate for specific actions of CDK2. Consequentially, $Cdk2^{-/-}$ meiocytes undergo meiotic arrest resulting in hypoplastic reproductive organs and infertility in both male and female animals (Berthet et al., 2003; Ortega et al., 2003).

In most mammals, CDK2 is expressed as two distinct protein isoforms, here referred to as CDK2S (short; 33 kDa) or CDK2L (long; 39 kDa) (Ellenrieder et al., 2001). The longer CDK2L transcript is generated by alternate splicing of the gene *CDK2*. In mice, this leads to the inclusion of exon 6 and, upon translation, the inclusion of this alternatively spliced exon results in a 48-amino-acid insertion. This alternate splicing event leads to the expression of CDK2L, which has been detected in many mammals (Baptist et al., 1996; Hain et al., 1994; Noguchi et al., 1993; Sweeney et al., 1996; Takano et al., 1994) and also in the monkey COS-1 cell line (Ellenrieder et al., 2001). The human *CDK2* gene contains seven exons, whereas mouse *Cdk2* contains eight exons (Ellenrieder et al., 2001; Kwon et al., 1998). The DNA segment encoding the 'extra' exon in the mouse *Cdk2* gene, exon 6, also exists in the human *CDK2* DNA sequence within intron 5. Although the mRNA transcript encoding the long isoform of murine *Cdk2* is listed as a putative transcript in humans (XP_011536034.1), this isoform is not expressed in several human cell lines. Additionally, it has been reported that CDK2L expression is not seen in human tissues such as thymus that are known to express both long and short *Cdk2* isoforms in mice (Ellenrieder et al., 2001; Liu et al., 2014; Sweeney et al., 1996).

The function of the 48-amino-acid insert encoded by exon 6 remains uncertain as its transcribed sequence does not share any homology with previously described protein domains. While there

[1]Department of Chromosome Biology, Max Perutz Labs, University of Vienna, Vienna Biocenter, Vienna 1030, Austria. [2]Department of Clinical Sciences, Lund University, Clinical Research Centre (CRC), Box 50332, SE-202 13 Malmö, Sweden. [3]Biozentrum der LMU München, Department Biologie II, Zell- und Entwicklungsbiologie, 82152 Planegg-Martinsried, Germany. [4]Institute of Molecular and Cell Biology (IMCB), A*STAR (Agency for Science, Technology and Research), 61 Biopolis Drive, Proteos, Singapore 138673, Republic of Singapore. [5]Center for OncoGenomics and Innovative Therapeutics (COGIT), Department of Oncological Sciences and Pharmacological Sciences, Tisch Cancer Institute, Icahn School of Medicine at Mount Sinai, New York, New York 10029, USA. [6]Department of Obstetrics and Gynecology, Li Ka Shing Faculty of Medicine, The University of Hong Kong, Hong Kong, China. [7]Shenzhen Key Laboratory of Fertility Regulation, Center of Assisted Reproduction and Embryology, The University of Hong Kong - Shenzhen Hospital, Haiyuan First Road 1, Shenzhen 518053, China.

*Author for correspondence ( philipp.kaldis@med.lu.se)

N.P., 0000-0002-1913-9454; N.E.K., 0000-0001-5895-2459; H.H., 0009-0007-4763-5782; U.K., 0000-0002-6771-3721; S.Z.A.T., 0000-0002-6063-254X; J.R.O., 0000-0002-7468-691X; T.T., 0000-0001-6873-3073; L.N.Z., 0000-0001-6552-0929; E.G., 0000-0001-7764-5307; K.L., 0000-0002-7519-3332; P.K., 0000-0002-7247-7591

have been few studies investigating the differences between the functions of CDK2L and CDK2S, evidence suggests that CDK2L could play a role during pachytene and may preferentially interact with Speedy A (SPDYA; see figure 1 in Tu et al., 2017), in addition to differences in activating and inhibitory phosphorylation (Bradley et al., 2023 preprint).

## RESULTS
### Generation of CDK2L and CDK2S mouse models
In an attempt to identify biological processes mediated specifically by CDK2L or CDK2S *in vivo*, we generated two different mouse strains. In the first, the deletion of exon 6 completely prevents the expression of the CDK2L, promoting the constitutive expression of only the shorter isoform CDK2S (Fig. 1, knockout). In the second, the fusion of exons 5, 6 and 7 removes the surrounding intronic sequences required for the skipping of exon 6, forcing the sole expression of CDK2L (Fig. 1, knock-in).

To confirm the correct expression of CDK2L or CDK2S in each of our mutant models, we performed western blotting of tissues extracted from homozygous CDK2L and CDK2S mice and compared the protein expression of CDK2 to that seen in wild type (Fig. 2). CDK2L is known to be highly expressed in mouse testes and thymus (Ellenrieder et al., 2001). In less proliferative tissues such as the brain, CDK2L cannot be detected at high levels. As previously described, CDK2 could be observed as two isoforms in wild-type testis, spleen and thymus, appearing at around 39 kDa

(CDK2L) and 33 kDa (CDK2S) but could not be detected in brain lysate (Fig. 2A). In extracts from CDK2L and CDK2S mice using a pan-CDK2 antibody that recognizes both isoforms (see lanes 1-3), only the long or short isoforms of CDK2 could be detected, respectively (Fig. 2A, lanes 5-7 and 9-11). Furthermore, these singular isoforms were expressed at levels similar to those seen for the combined expression of *Cdk2* isoforms expressed in wild-type tissues, as can be seen in testis extracts (Fig. 2B). Together, these results suggest that the CDK2L and CDK2S mouse models reported in this study correctly express only the long and short isoforms of CDK2, respectively, and that this expression occurs at similar levels to that seen for wild-type CDK2 in various tissues. Phenotypically, homozygous CDK2L and CDK2S mice were found to be essentially indistinguishable from wild-type mice aside from a small decrease in body weight during early development for CDK2L (see Fig. 4A).

To confirm the orderly development of reproductive organs, we measured the weight of the testis and prepared Haematoxylin and Eosin (H&E)-stained sections from *Cdk2S* and *Cdk2L* testis and ovary. Although the bodyweight and testis weight were slightly decreased in *Cdk2L* mice, the ratio of testis/bodyweight was not significantly reduced (see Fig. 4A). In postnatal day (P) 30 testes of *Cdk2S* and *Cdk2L* mice, we observed a comparable appearance of metaphase I stage primary spermatocytes, round spermatids and elongating spermatids, indicating the correct progression of spermatocytes through both meiotic divisions (Fig. 3A and data not shown). For

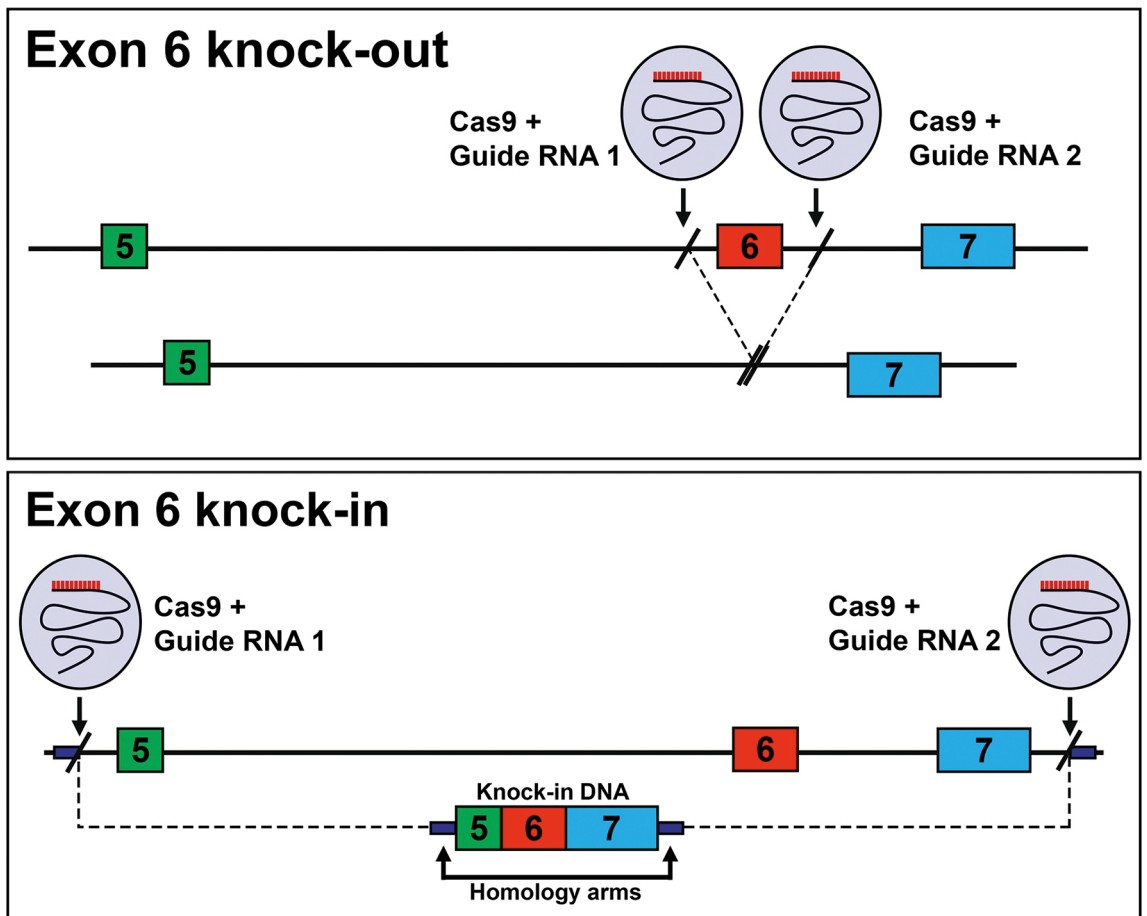

**Fig. 1. Design of the CDK2S and CDK2L mice.** Top: Deletion of exon 6 leads to mice that can only express CDK2S (Exon 6 knock-out; *Cdk2^SHORT/SHORT^*). Bottom: Fusion of exons 5-7 leads to the continuous expression of CDK2L and the absence of CDK2S (Exon 6 knock-in; *Cdk2^LONG/LONG^*).

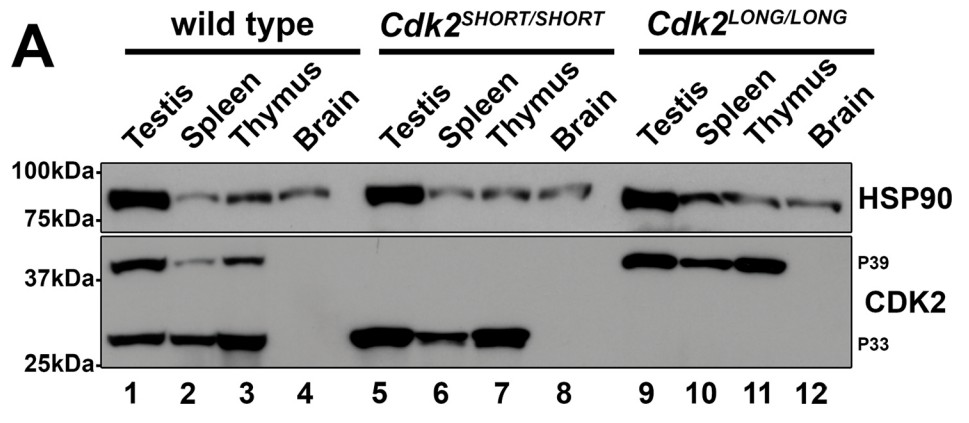

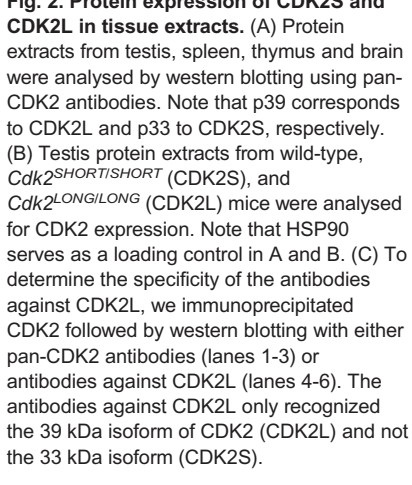

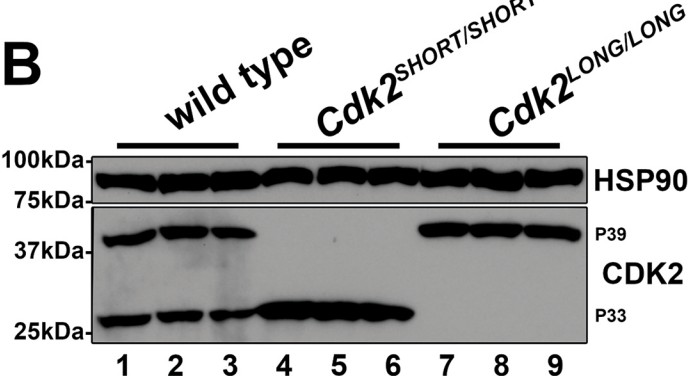

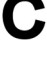

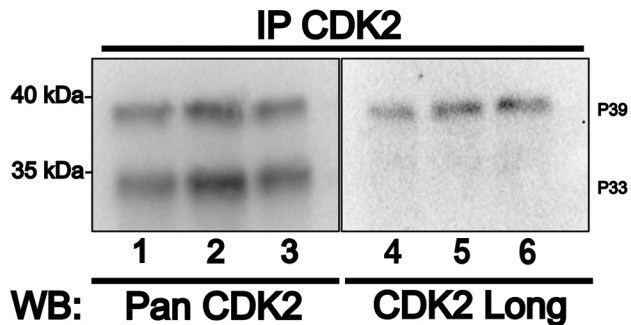

**Fig. 2. Protein expression of CDK2S and CDK2L in tissue extracts.** (A) Protein extracts from testis, spleen, thymus and brain were analysed by western blotting using pan-CDK2 antibodies. Note that p39 corresponds to CDK2L and p33 to CDK2S, respectively. (B) Testis protein extracts from wild-type, *Cdk2^{SHORT/SHORT}* (CDK2S), and *Cdk2^{LONG/LONG}* (CDK2L) mice were analysed for CDK2 expression. Note that HSP90 serves as a loading control in A and B. (C) To determine the specificity of the antibodies against CDK2L, we immunoprecipitated CDK2 followed by western blotting with either pan-CDK2 antibodies (lanes 1-3) or antibodies against CDK2L (lanes 4-6). The antibodies against CDK2L only recognized the 39 kDa isoform of CDK2 (CDK2L) and not the 33 kDa isoform (CDK2S).

female mice, ovarian follicles were observed to develop similarly in adult (P56) CDK2L, CDK2S and wild-type mice (Fig. 3B) and the overall histology of the ovaries did not show any significant differences between the genotypes. Together, these results demonstrate that the reproductive organs of CDK2S and CDK2L mice are histologically indistinguishable from wild type, suggesting that these mice could be fertile.

Both male and female *Cdk2^{−/−}* (lacking both CDK2S and CDK2L) mice are viable but show sterility at 100% penetrance in crosses between adults (8-12 weeks old; Berthet et al., 2003). Based on this information, it was surprising that heterozygous or homozygous mutant intercrosses of mice led to viable pregnancies in both CDK2L and CDK2S genotypes. For all combinations of the intercrosses performed, the average litter size was not significantly

different from crosses of wild type and expected Mendelian ratios were observed (Fig. 4B). To confirm that fertility is sustained into adulthood, we followed breedings for at least five litters and found that there was no obvious decrease in fecundity over this time and mutant mice continued to have comparable numbers of pups to wild type at intervals that were not significantly longer than those of wild type. The gender of the born pups was almost equally divided between female and male, and this was not dependent on the genotype of the parents (Fig. 4C-F). Finally, we analysed heterozygote crosses and found the expected 25% (+/+), 50% (+/−) and 25% (−/−) Mendelian ratios of the offspring (Fig. 4G,H). These data demonstrate that both CDK2S and CDK2L mice are fully fertile, which indicates that both CDK2 isoforms are able to drive meiotic development *in vivo*.

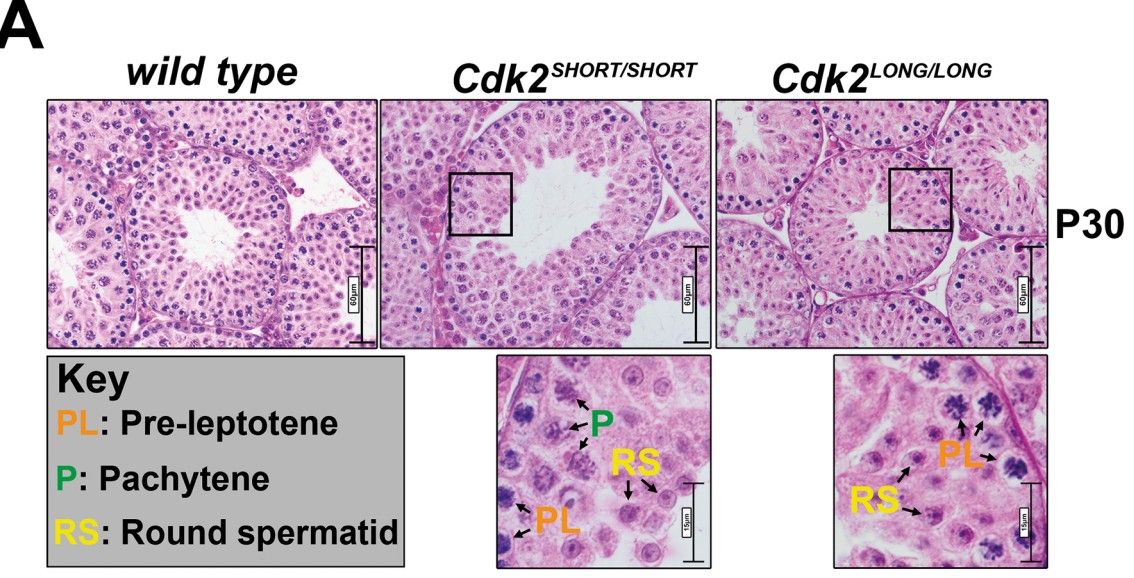

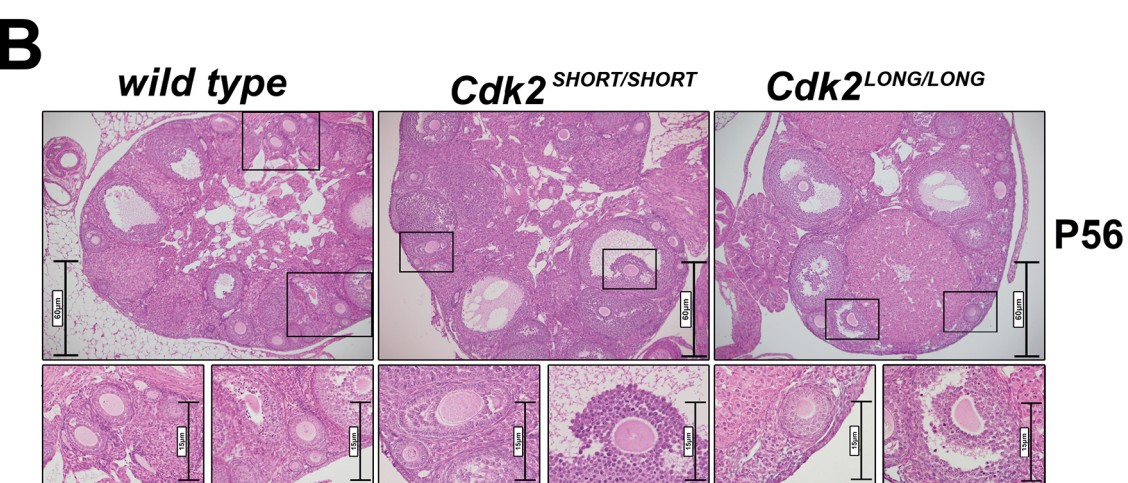

**Fig. 3. Morphology of testis and ovary.** (A) H&E-stained P30 (30 days postpartum) testis sections from wild type, *Cdk2^SHORT/SHORT* (CDK2S) and *Cdk2^LONG/LONG* (CDK2L). Boxed areas are shown at higher magnification beneath. Arrows indicate pre-leptotene (PL; orange), pachytene (P; green), spermatocytes, and round spermatids (RS; yellow). Scale bars: 60 μm (top row); 15 μm (bottom row). (B) H&E-stained P56 ovary sections from wild type, *Cdk2^SHORT/SHORT* (CDK2S) and *Cdk2^LONG/LONG* (CDK2L). Boxed areas are shown at higher magnification beneath to illustrate follicles. Scale bars: 60 μm (top row); 15 μm (bottom row).

## CDK2L localizes specifically to telomeres during meiotic prophase I but can be functionally replaced by CDK2S to ensure homologue pairing

The comparable reproductive health of CDK2L and CDK2S mutant mouse models suggests that these isoforms are functionally redundant with regard to maintaining fertility. We hypothesized that both of these isoforms must be catalytically active and able to act upon the same substrates to maintain successful meiotic division.

In mammalian meiocytes, CDK2 can be detected cytologically at telomeres throughout meiotic prophase and transiently, at late recombination nodules, specifically during mid-pachytene (Ashley et al., 2001; Palmer et al., 2020). Telomeric CDK2 is known to mediate telomeric tethering as part of the LINC (linker of nucleoplasm and cytoskeleton) complex (Viera et al., 2015, 2009; for a review, see Link et al., 2015). In contrast, late recombination nodule-associated CDK2 orchestrates the maturation sites of meiotic recombination to form meiotic crossovers (Palmer et al.,

2020). Although not currently associated with specific binding on chromosomes, further evidence also suggests that CDK2 might also act to control transcription during meiotic prophase I (Palmer et al., 2019b). Upon deletion of *Cdk2* or in knock-in models in which *Cdk2* is catalytically inactive, the activity of *Cdk2* at telomeres is lost. In these instances, meiotic arrest occurs as a result of the failure of homologue pairing during the zygotene-pachytene transition of meiotic prophase I (Chauhan et al., 2016; Viera et al., 2015, 2009). In contrast, in knock-in models of *Cdk2*, in which the action of *Cdk2* at telomeres is not perturbed and homologue pairing occurs successfully, telomere fusion events increase, leading to apoptosis for several reasons (Chauhan et al., 2016; Palmer et al., 2020).

To determine whether any meiotic defects occur following the loss of either the long or short isoforms of *Cdk2*, we prepared chromosome spreads from CDK2L and CDK2S spermatocytes. First, we performed dual staining for the lateral and transverse elements of the synaptonemal complex scaffolding complex, SYCP3 and SYCP1. The colocalization of SYCP3 and SYCP1

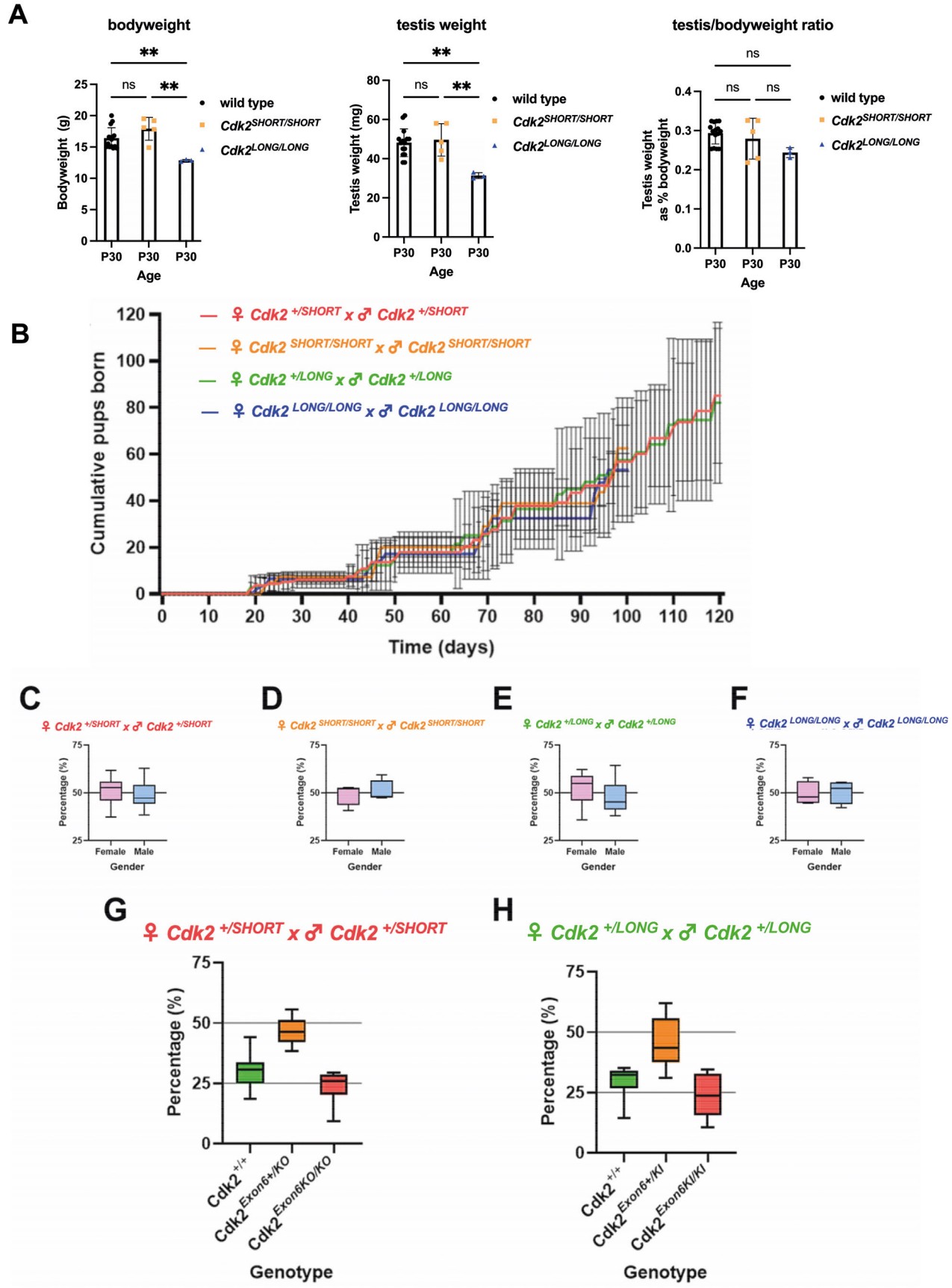

**Fig. 4.** See next page for legend.

**Fig. 4. CDK2S and CDK2L mice are fertile.** (A) P30 body weight, testis weight, and testis weight as a percentage of bodyweight for wild type (black circles, n=15), $Cdk2^{SHORT/SHORT}$ (CDK2S; orange squares, n=5) and $Cdk2^{LONG/LONG}$ (CDK2L, blue triangles, n=3). *P<0.05 and **P<0.01 (one-way ANOVA with Tukey's multiple comparison post-hoc test). ns. not significant. Bars and error bars represent mean and s.d., respectively. (B) Cumulative pups born over time for crosses between $Cdk2^{+/SHORT}$ (red; N=9 breeding pairs), $Cdk2^{SHORT/SHORT}$ (orange; N=4 breeding pairs), $Cdk2^{+/LONG}$ (green; N=8 breeding pairs) or $Cdk2^{LONG/LONG}$ (blue; N=8 breeding pairs). Data are presented as the mean number of cumulative pups born to breeding pairs of each condition (±s.d.). One-way ANOVA analysis indicated that there is no statistical significance between heterozygous groups. Day 0 indicates the start of breeding, which was initiated when both male and female mice of a single breeding pair were 8-10 weeks of age. (C-F) Gender balance of the pups from the breedings shown in B. (G) Percentage of wild-type ($Cdk2^{+/+}$), heterozygote ($Cdk2^{+/SHORT}$) and homozygote ($Cdk2^{SHORT/SHORT}$) pups from heterozygote intercrosses. (H) Percentage of wild-type ($Cdk2^{+/+}$), heterozygote ($Cdk2^{+/LONG}$) and homozygote ($Cdk2^{LONG/LONG}$) pups from heterozygote intercrosses. Note that the expected 25%, 50%, 25% Mendelian ratios for genotype, and 50% for male and female ratios were observed, tested by $\chi^2$ goodness of fit test. Percentages in graphs in C-H were calculated based on the number of pups born for given traits (e.g. gender or genotype) in a breeding pair. Boxes represent the interquartile range of the percentage of pups per breeding pair (n=8 for heterozygous and n=4 for homozygous). The median is indicated. The whiskers show values extending to 1.5× interquartile range from the quartile 1 and quartile 3 boundaries.

during meiotic prophase and the formation of 19 well-separated bivalents in addition to the XY pair indicated successful synapsis in both CDK2S and CDK2L testes (Fig. 5). As synapsis was comparable in CDK2L and CDK2S mice, we infer that these two isoforms of CDK2 are functionally redundant and can each act independently of the other to maintain the meiotic functions of CDK2 regarding synapsis.

### Repair of recombination intermediates during meiotic prophase I

During meiotic prophase I, disruption of double-strand break repair and failure of homologue pairing are believed to trigger apoptosis and arrest in spermatocytes. A useful marker to monitor repair of DNA damage is the phosphorylated histone H2A variant (γH2AX; Ser139). We observed that the γH2AX signal in wild-type spermatocytes is 'cloud-like' in leptotene and zygotene (Fig. 6A,B), localized along unpaired axes in early-pachytene (Fig. 6C), which progressively clears up between mid-pachytene and diplotene (Fig. 6D-F). The sex chromosomes remained stained by a cloud of γH2AX during all stages (Fig. 6A-F). In both $Cdk2^{SHORT/SHORT}$ and $Cdk2^{LONG/LONG}$ mice, which express only CDK2S and CDK2L, respectively, the pattern of γH2AX mirrored that of wild-type mice (Fig. 6G-R). Quantification of the total H2AX fluorescence (Fig. 6S-X) as well as the overlap of the H2AX/SYCP3 fluorescence (Fig. 6Y,Z) indicate minor differences with the exception of CDK2L (Fig. 6Y). Therefore, we conclude that repair of recombination intermediates due to strand invasion is effective in the mutant mice and that early stages of meiotic prophase are not majorly affected by the lack of either CDK2S or CDK2L.

### Both CDK2L and CDK2S can associate with Speedy A at the nuclear envelope to maintain homologue pairing

The multifunctional nature of CDK2 during meiotic prophase likely arises from the association of CDK2 with a variety of different partner proteins, including cyclins, cyclin-like/atypical cyclin proteins (Quandt et al., 2020) and non-cyclins. To date, the contribution of these CDK2-binding partners towards the various actions of CDK2 during meiotic prophase is poorly understood (for a review, see Palmer et al., 2019a). Currently, one of the most well-characterized meiosis-specific CDK2 partners is the cyclin-like

protein, Speedy/Ringo A (Ferby et al., 1999; Lenormand et al., 1999; Porter et al., 2002). Like CDK2, Speedy A is required for the pairing of homologous chromosomes and its deletion results in meiotic arrest essentially indistinguishable from that seen upon $Cdk2$ deletion (Mikolcevic et al., 2016; Tu et al., 2017). Speedy A has been proposed to promote the loading of CDK2 onto telomeres (Tu et al., 2017). Similarly, the deletion of CDK2 leads to a loss of Speedy A binding at telomeres (Palmer et al., 2020). An added layer of complexity is the observation that the E-type cyclins, especially cyclin E2, are required to maintain the integrity of telomeres during meiotic prophase I (Manterola et al., 2016; Martinerie et al., 2014). Deletion of cyclin E2 causes a loss of telomere integrity, which also results in pairing defects. This is worsened by the concurrent loss of cyclin E1, indicating that both E-type cyclins, in addition to Speedy A, are required for homologue pairing during meiotic prophase. Interestingly, the additional 48-amino-acid insertion seen in the CDK2L protein is positioned almost directly opposite the cyclin-binding face of CDK2. This location is distant from the cyclin interface, the ATP-binding sites and the CKS-binding sites (Ellenrieder et al., 2001). One suggested possibility is that this sequence might influence the interaction between CDK2L and potential binding partners in an indirect way.

CDK2L is known to be expressed preferentially in both male and female meiotic cells, specifically during the timing of meiotic recombination between homologous chromosomes (Tu et al., 2017; Wang et al., 2014). The relative expression CDK2S and CDK2L in testis also changes during development. CDK2S is detectable starting from P8 until the end of pachytene and is almost undetectable in round and elongated spermatids (see figure 1B,C in Tu et al., 2017). In contrast, CDK2L appears at P12 and is expressed from then onwards, concurrently with CDK2S. Interestingly, Speedy A starts to be expressed at the same time as CDK2L at P12 (see figure 1C in Tu et al., 2017). The localization of CDK2L to telomeres seem to be dependent on binding to Speedy A (see figure 8C-L in Tu et al., 2017) and probably on CDK2L activity. Similarly to CDK2L, during meiosis, Speedy A can be detected specifically at telomeres and along the sex body (Mikolcevic et al., 2016; Palmer et al., 2020; Tu et al., 2017) but not at late recombination nodules. At the telomeres, Speedy A has been implicated in recruiting telomere-bound CDK2 to the nuclear envelope to stabilize telomere-nuclear envelope tethering (Tu et al., 2017).

Based on this previous knowledge, we stained chromosome spreads with antibodies against Speedy A and pan-CDK2 to detect all isoforms of CDK2. In wild-type mice, we found that Speedy A localizes exclusively to telomeres, whereas CDK2 binds to both telomeres and interstitial sites, as expected (Fig. 7, I-IV). At the telomeres, Speedy A and CDK2 were colocalized. In the absence of CDK2L ($Cdk2^{SHORT/SHORT}$), Speedy A binds to telomeres, as in the wild type, and colocalized with CDK2S at the telomeres but not at the interstitial sites (Fig. 7, V-VIII). In the absence of CDK2S in the $Cdk2^{LONG/LONG}$ mice, Speedy A still localized to the telomeres, now overlapping with CDK2L (Fig. 7, IX-XII). Therefore, in all three genotypes Speedy A localization was identical at the telomeres and was thus able to interact with both isoforms of CDK2. Importantly though, the binding partner of CDK2 at the interstitial sites is still unknown.

### Specific localization of CDK2 in $Cdk2^{SHORT/SHORT}$ and $Cdk2^{LONG/LONG}$ mice

As previous studies investigating CDK2 localization during meiosis were carried out using antibodies detecting both isoforms of CDK2 (Ashley et al., 2001; Chauhan et al., 2016; Palmer et al., 2020), we

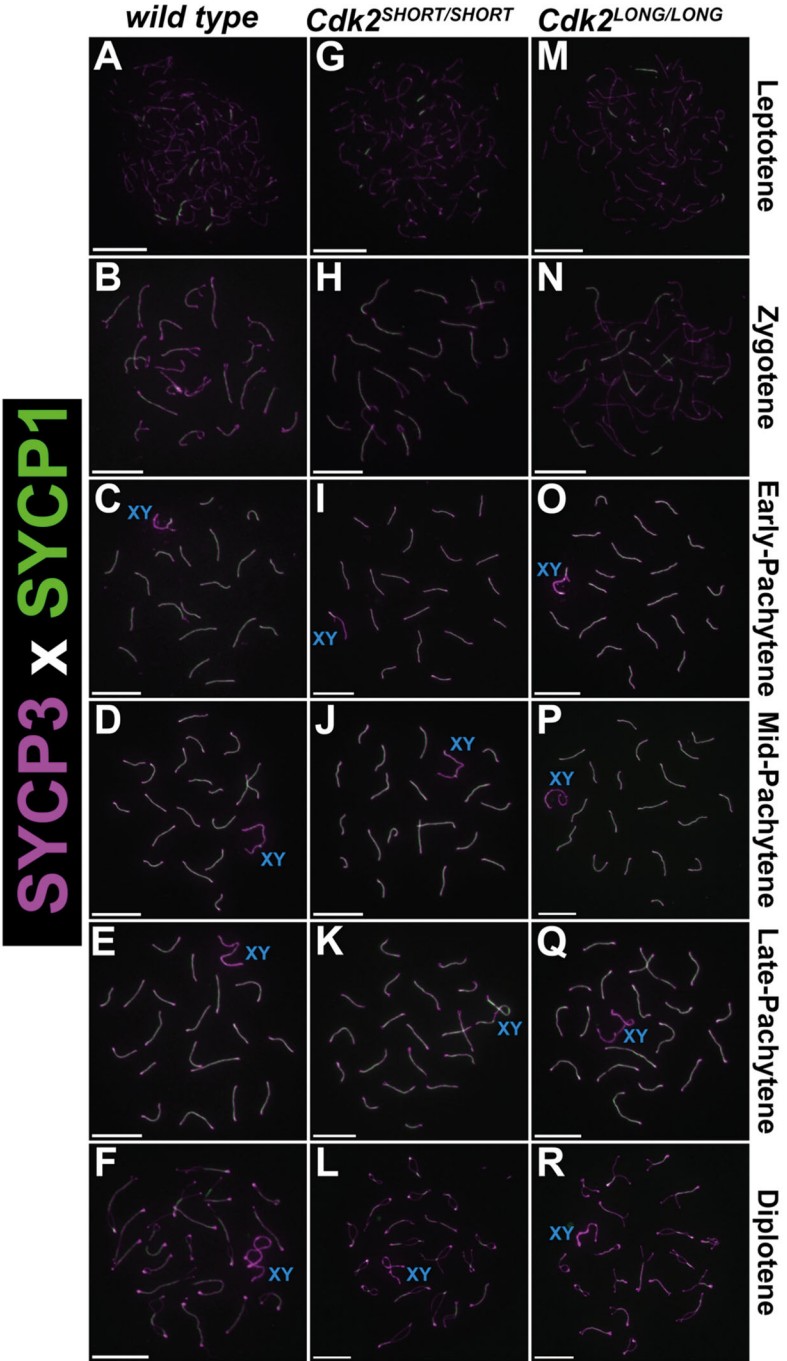

**Fig. 5. The synaptonemal complex forms successfully in CDK2S and CDK2L testis.** (A-R) Chromosome spreads were prepared from adult testis and stained with antibodies against SYCP1 (green) and SYCP3 (magenta). Various stages of meiotic prophase I are depicted for wild type (A-F), *Cdk2^{SHORT/SHORT}* (G-L; CDK2S) and *Cdk2^{LONG/LONG}* (M-R; CDK2L). Spermatocytes from all three genotypes progress comparably into mid-pachytene (D,J,P), late-pachytene (E,K,Q) and diplotene (F,L,R). The sex chromosomes are labelled by a blue 'XY'. At least three biological replicates were analysed for each condition. Scale bars: 5 µm.

designed antibodies to target CDK2L specifically in an attempt to determine whether this isoform displayed specific temporal or special expression during meiotic prophase I (Fig. 2C; for a detailed description, see Materials and Methods). In wild-type mice, we found that detection of CDK2 with antibodies specifically against CDK2L revealed CDK2 at telomeres but not interstitial sites marking locations of meiotic recombination (Fig. 8, I-IV, red). In contrast, pan-CDK2 antibodies, which detect both the long and short forms of CDK2, stained both telomeres and interstitial sites marking locations of meiotic recombination (Fig. 8, I-IV, green). These data indicate that CDK2L has a higher affinity for binding at telomeres compared to interstitial sites. As it is not possible to design specific antibodies against CDK2S that do not cross-react with CDK2L, we determined the binding of CDK2 in our CDK2S

mouse model. In the absence of CDK2L (*Cdk2^{SHORT/SHORT}*), CDK2S is the only isoform of CDK2 expressed and binds both to the telomeres and the interstitial sites (Fig. 8, V-VIII, green). This suggests that CDK2S is able to bind to both sites, at least in the absence of CDK2L. Based on this finding, it was important to check where CDK2L binds in the absence of CDK2S in the *Cdk2^{LONG/LONG}* mice. Interestingly, CDK2L was localized to both telomeres and interstitial sites, albeit with a stronger signal at the telomeres (Fig. 8, IX-XII, red). This suggests that CDK2S and CDK2L can compensate for each other, especially in the absence of the other isoform, which is similar to the compensation seen between different CDKs (Aleem et al., 2005; Berthet et al., 2003; Berthet and Kaldis, 2007; Kaldis and Aleem, 2005; Satyanarayana and Kaldis, 2009).

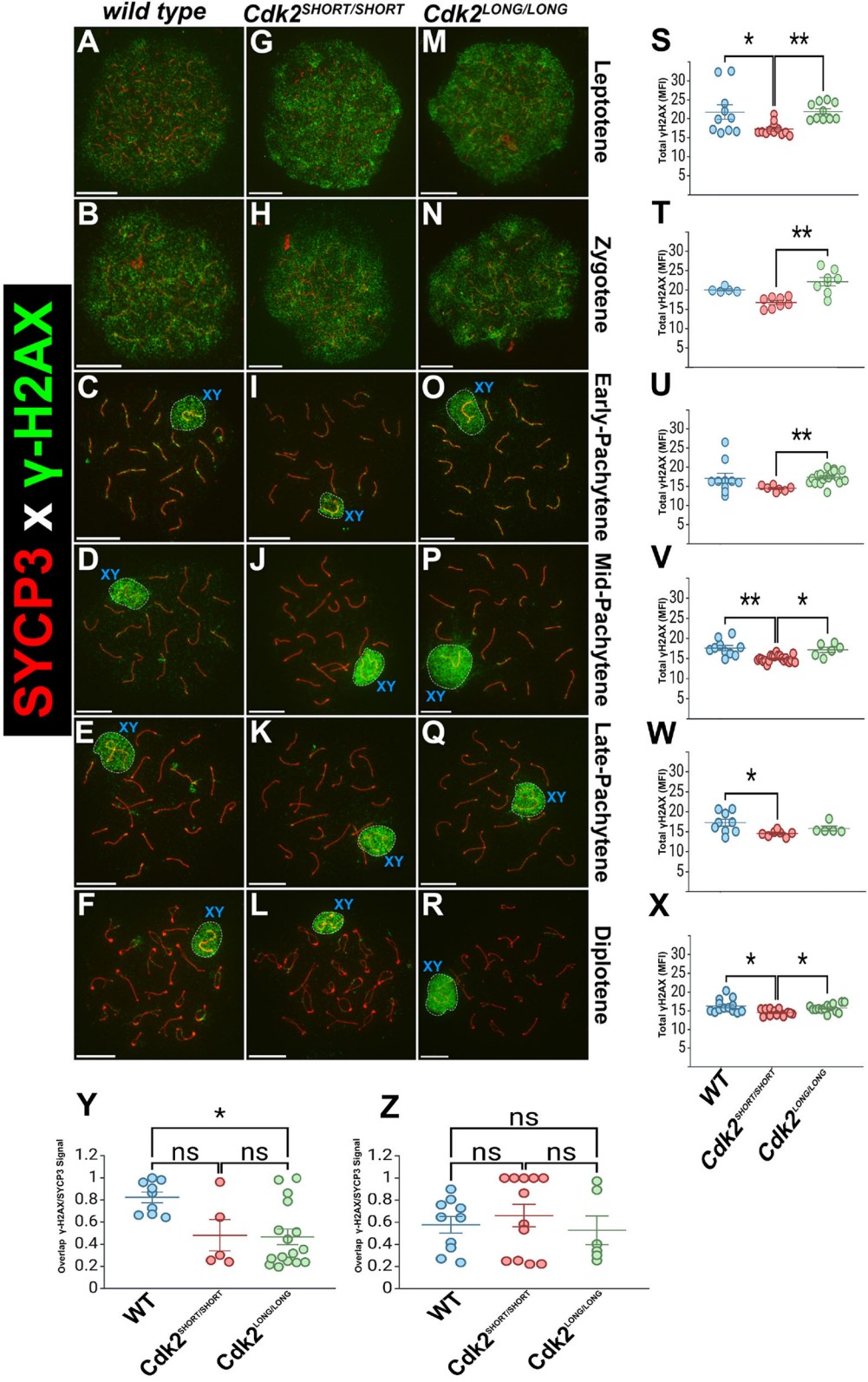

**Fig. 6. The repair of recombination intermediates.** Chromosome spreads were prepared from adult testis and stained with antibodies against γH2AX (green, DNA damage) and SYCP3 (red). Various stages of meiotic prophase I are depicted for wild type (A-F), *Cdk2*<sup>SHORT/SHORT</sup> (G-L; CDK2S) and *Cdk2*<sup>LONG/LONG</sup> (M-R; CDK2L). DNA intermediates are repaired at the correct time with a cloud-like staining in leptotene (A,G,M) and zygotene (B,H,N). Thereafter, γH2AX staining is mostly found along the axis (C,I,O) and retreats over time starting at mid-pachytene (D,J,P,E,K,Q,F,L,R). The sex chromosomes are labelled by a blue 'XY'. At least three biological replicates were analysed for each condition. Scale bars: 5 µm. (S-Z) Statistics for overall H2AX fluorescence intensity for each stage (S-X) and overlap of H2AX and SYCP3 fluorescence intensity for early-pachytene (Y) and mid-pachytene (Z). *$P<0.05$, **$P<0.01$. Error bars show mean and s.e.m. Details of the statistical analysis are described in the Materials and Methods. ns, not significant. MFI, mean fluorescence intensity

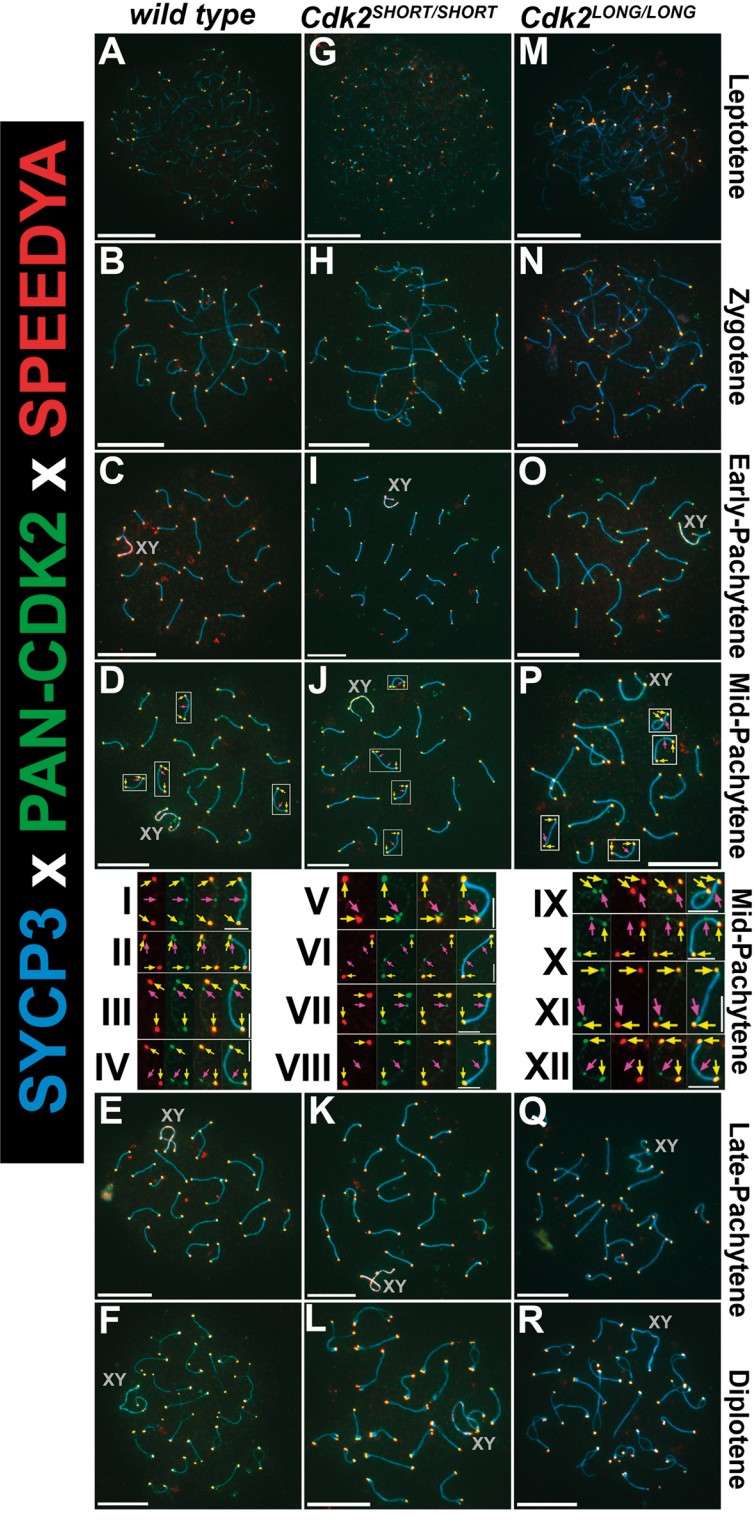

**Fig. 7. Speedy A can interact with both CDK2S and CDK2L.**
(A-R) Chromosome spreads from adult testis stained with antibodies against Speedy A (red), pan-CDK2 (green) and SYCP3 (blue). Various stages of meiotic prophase I are depicted for wild type (A-F), *Cdk2^{SHORT/SHORT}* (G-L; CDK2S) and *Cdk2^{LONG/LONG}* (M-R; CDK2L). Speedy A localizes to the telomeres in all genotypes, i.e. wild type (both CDK2S and CDK2L), *Cdk2^{SHORT/SHORT}* or *Cdk2^{LONG/LONG}*. The sex chromosomes are labelled by a grey 'XY'. Boxed areas are shown at higher magnification in insets I-XII. Yellow arrows indicate telomeres, and magenta arrows interstitial sites. At least three biological replicates were analysed for each condition. Scale bars: 5 µm (main panels); 1.25 µm (insets, I-XII).

These results suggest that CDK2L exhibits preferential binding to telomeres but not late recombination nodules during meiotic prophase in wild-type mice. Upon sole expression of CDK2S, however, the shorter CDK2 isoform can also effectively bind to telomeres, presumably replacing the function of CDK2L. Conversely, upon the sole expression of CDK2L, the longer CDK2 isoform can bind to late recombination nodules in addition to telomeres.

An interesting possibility here is that there exist distinct CDK2-binding proteins at telomeres and late recombination nodules that

show preferential binding of CDK2L and CDK2S, respectively. If the preferred CDK2 isoform is absent, however, the remaining pool of CDK2 would be recruited to these sites.

## DISCUSSION

Based on accumulated data from more than 20 years, we set out to investigate the *in vivo* functions of the two isoforms of cyclin-dependent kinase 2 (CDK2), CDK2S and CDK2L. In most studies, CDK2S has been investigated but there are a few studies on

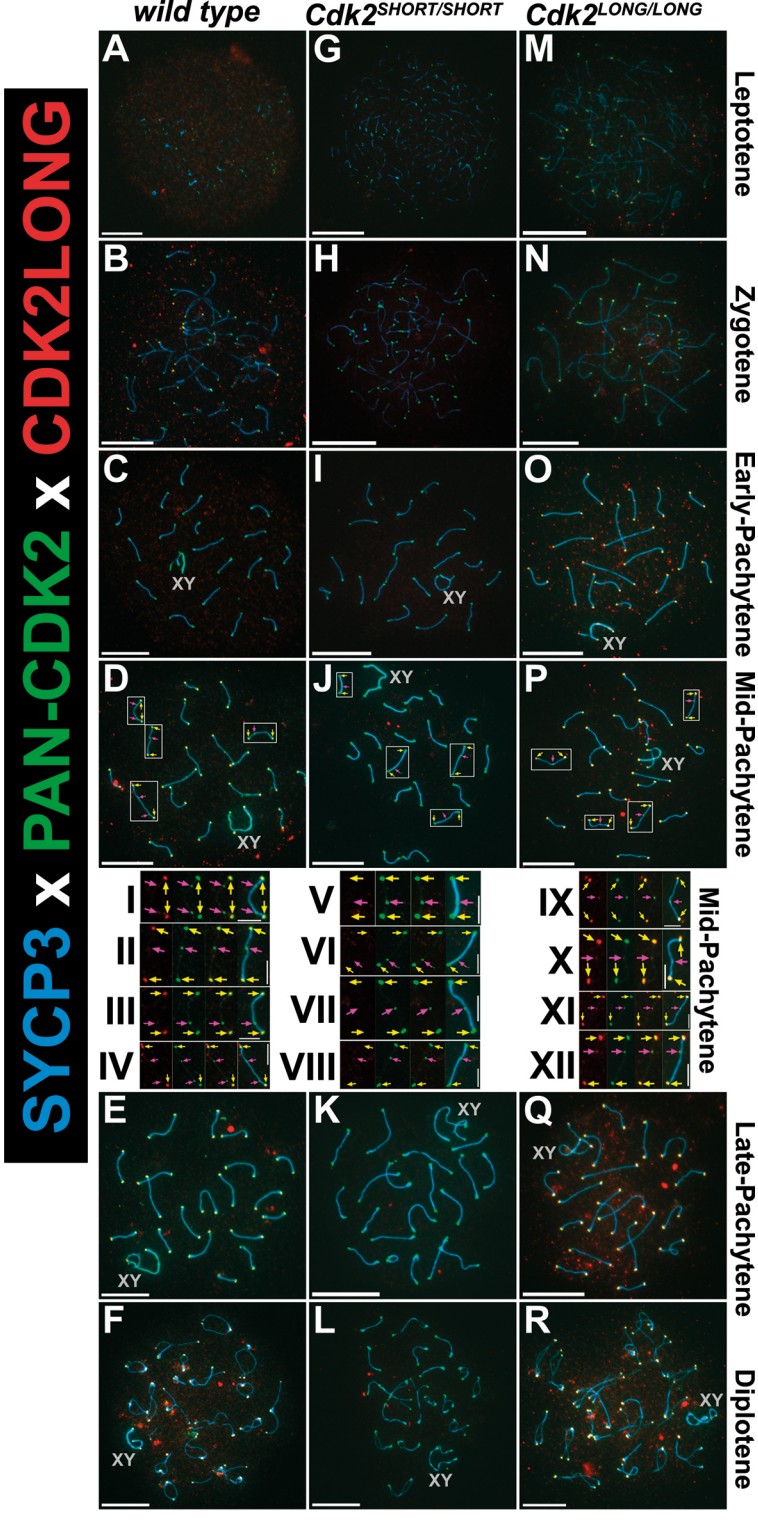

**Fig. 8. CDK2L can localize to interstitial sites in the absence of CDK2S.** (A-R) Chromosome spreads were prepared from adult testis for wild type (A-F), *Cdk2^{SHORT/SHORT}* (G-L; CDK2S) and *Cdk2^{LONG/LONG}* (M-R; CDK2L) and stained with antibodies against CDK2L (red), pan-CDK2 (green; detecting both CDK2S and CDK2L) and SYCP3 (blue). CDK2S can bind to both telomeres and interstitial sites (I-XII; green). In wild type, CDK2L binds to telomeres but not interstitial sites (I-IV; red); in *Cdk2^{SHORT/SHORT}* CDK2L cannot be detected (V-VIII); and in *Cdk2^{LONG/LONG}* CDK2L binds prominently to telomeres and weakly to interstitial sites (IX-XII; red). The sex chromosomes are labelled by a grey 'XY'. Boxed areas are shown at higher magnification in insets I-XII. Yellow arrows indicate telomeres, and magenta arrows interstitial sites. At least three biological replicates were analysed for each condition. Scale bars: 5 µm (main panels); 1.25 µm (insets, I-XII).

CDK2L. In order to achieve this, we generated mouse strains that only expressed one of the isoforms but not the other. Using these mice, we studied fertility, which is the major phenotype in *Cdk2^{null}* mice (Berthet et al., 2003; Ortega et al., 2003). Although there might be biochemical differences in CDK2S and CDK2L *in vitro*, we found that *in vivo* these two isoforms can compensate for each other. This resulted in two mouse stains, CDK2S and CDK2L, that are viable and fully fertile, despite that CDK2L and CDK2S display differences in kinase activity *in vitro*. This may reflect the essential

role of CDK2 during the meiotic cell cycle, and therefore may imply that the gene function does not exclusively depend on enzyme activity, but also on other factors such as substrate accessibility and subcellular localization; however, more studies will need to be performed in that regard.

One of the interesting and surprising observations was that, although both mouse strains are viable, CDK2L mice seem to be slightly smaller than wild-type and CDK2S mice. Originally, we and others found that *Cdk2^{null}* mice are consistently 10-15% smaller

than wild-type mice (Berthet et al., 2003; Ortega et al., 2003) but we could not uncover the reasons behind this phenotype. From our current data, it seems that the body weight phenotype is associated either with the expression of CDK2L or the lack of CDK2S. The testis weight followed the same pattern, i.e. decreased in the CDK2L mice, in accordance with the overall body weight data. Since this is a mild phenotype, determining the molecular reason behind the body size decrease will be challenging.

The majority of the efforts reported in this study is dedicated to spermatogenesis, in which CDK2 plays multiple roles (Palmer and Kaldis, 2020). Although it became quickly clear that CDK2S and CDK2L mice were fertile, we investigated several stages of meiosis and all the known functions of CDK2 but ultimately had to conclude that CDK2S and CDK2L can compensate for each other's functions at least in the laboratory setting. Interestingly, the specific antibodies against CDK2L that we generated stained CDK2L at the telomeres very well, as expected, but the staining of the interstitial sites was low to undetectable. This confirmed that CDK2L has a higher affinity for the telomeres but there was still CDK2 localized to the interstitial sites in the absence of CDK2S (Fig. 8, IX-XII). The importance of these findings lies in the specificity of our antibodies against CDK2L (Fig. 2C; see also Materials and Methods), which detect CDK2L but not CDK2S (Fig. 8, V-VIII). We do not believe this is an effect of affinity of the CDK2L antibodies, but we cannot exclude the remote possibility that CDK2 is part of a complex at the interstitial sites, which block accessibility of the CDK2L antibodies but not of the pan-CDK2 antibodies. Therefore, we conclude that CDK2L is preferentially localized to telomeres but retains some expression at the interstitial sites. The functional consequences of the localization of CDK2L to telomeres needs to be further investigated in the future.

Overall, we have found that *in vivo* both isoforms can perform all the functions that CDK2 is important for in mice in the laboratory setting. Unfortunately, our work cannot explain why the splice isoforms of CDK2 are conserved in several mammals but not in humans.

The limitations of our study are that, despite investing a lot of efforts in probing the known functions of CDK2, there may be other conditions that would have shown a clear difference between the CDK2S and CDK2L mice. Such conditions could include ageing, DNA damage in various contexts such as irradiation, or crossing the mice to $Cdk4^{null}$ or $p27^{Kip1}$ mice as we previously did with $Cdk2^{null}$ mice (Aleem et al., 2005; Berthet et al., 2006) to determine the compensation by CDK1 and CDK4. Further extensions could include investigation of the decrease in testis weight in CDK2L by analysing apoptosis and staining for MLH1/3 in chromosome spreads to determine the number of crossovers. Nevertheless, our studies have revealed that CDK2S and CDK2L have overlapping functions and loss of either isoform leads to fertile offspring, which addresses a long-standing question in the cell cycle/CDK2 research field.

## MATERIALS AND METHODS
### Animal studies
All animal experiments were carried in accordance with the Animal Care and Use Committee of Biological Resource Centre at Biopolis, A*STAR, Singapore (protocol #171268) and Swedish laws and regulations on animal experimentation. All mice were kept under 12 h light and dark cycles. The animals were fed with *ad libitum* chow diet and water. The wild-type C57BL/6 mice were acquired form The Jackson Laboratory. The mice were euthanized by cervical dislocation. $Cdk2^{Exon6KO}$ and $Cdk2^{Exon6KI}$ mice were generated by Cyagen Biosciences (Guangzhou).

### Generation of $Cdk2^{LONG}$ mice
The gRNA for the mouse *Cdk2* gene, the donor vector containing the *Cdk2* gene exon 6 cassette, and *Cas9* mRNA were co-injected into fertilized mouse eggs to generate targeted knock-in offspring. F0 founder animals were identified by PCR followed by sequence analysis, which were bred to wild-type mice to test germline transmission and F1 animal generation. gRNAs used for generation of $Cdk2^{LONG}$ mice were: gRNA1 (matching forward strand of gene), AAATGGTATGGAGGCTTGCCAGG; gRNA2 (matching forward strand of gene), CCCATTTCCAGGTGACCCGCAGG; gRNA3 (matching forward strand of gene), TCTTTGCTGAAATGGT-ATGGAGG; gRNA4 (matching reverse strand of gene), ATAGGGC-CCTGCGGGTCACCTGG. The *Cdk2* gene exon 6 cassette sequence (homology arms are underlined) was: AGTACTACTCCACAGCCGTGG-ATATCTGGAGCCTGGGCTGCATCTTTGCTGAAATGCACCTAGTG-TGTACCCAGCACCATGCTAAGTGCTGTGGGGAACACAGAAGAA-ATGGAAGACACAGTCTCTGCCCGCTGTGCTCCTATCTAGAAGTG-GCTGCATCACAAGGAGGGGGGGATGACCGCAGTGTCTGCCCCAC-ACCCCGTGACCCGCAGGGCCCTATTCCCTGGAGATTCTGAGATT-GACCAACTCTT.

### Generation of $Cdk2^{SHORT}$ mice
The gRNA for the mouse *Cdk2* gene, and *Cas9* mRNA were co-injected into fertilized mouse eggs to generate targeted knockout offspring. F0 founder animals were identified by PCR followed by sequence analysis, which were bred to wild-type mice to test germline transmission and F1 animal generation. gRNAs used for generation of $Cdk2^{SHORT}$ mice were: gRNA1 (matches reverse strand of gene), GTTTCGAATAAGAGGTCTATAGG; gRNA2 (matches forward strand of gene), ACCGACCCCATGATAAGCCCTGG.

### Histology
For H&E staining, testes and ovaries were isolated at the indicated time points and fixed in modified Davidson's solution, then transferred into 70% ethanol, embedded in paraffin, and cut into 6 mm sagittal sections with a microtome. To evaluate proliferation in the ovary, histological sections were stained using Ki-67 antibodies (Leica Microsystems, NCL-Ki67p). Images were captured using either an Olympus BX61 or Zeiss Imager Z1 microscope.

### Generation of antibodies against CDK2L
Specific antibodies against CDK2L were raised in rabbits using a peptide corresponding to the sequence encoded by mouse *Cdk2* exon 6 (CPLCSYLEVAASQGGGMTAVS), which is not part of the CDK2S protein sequence. Peptides were synthesized by GL Biochem (Shanghai) Ltd. and coupled to keyhole limpet haemocyanin (KLH) with the crosslinker disuccinimidyl suberate, as has been described (Berthet et al., 2003). One milligram of peptide was injected subcutaneously into rabbits at days 0, 14, 35, 56, 77, 98, 119, 140, 161, 182, 203 and 224. Rabbits were bled at days 49, 70, 91, 112, 133, 154, 175, 196, 217 and 238 and sera were stored at 20°C. The specificity of the serum was tested by western blotting (Fig. 2C) and immunofluorescence on chromosome spreads (Fig. 8).

### Western blots
Preparation of protein extracts and western blotting procedures were carried out as described (Palmer et al., 2019b). For the validation of the antibodies against CDK2L, we immunoprecipitated CDK2 (both CDK2S and CDK2L) using 4 µl of mouse monoclonal anti-Cdk2 antibody (Santa Cruz Biotechnology, sc-6248) and 10 µl Protein A agarose slurry (MedChemExpress, HY-K0213) from mouse testis extract, followed by western blotting with rabbit anti-CDK2 antibodies (Berthet et al., 2003) at 1:5000 (Fig. 2, lanes 1-3) or rabbit antibodies against CDK2L (Berthet et al., 2003) at 1:1000, followed by a secondary anti-rabbit IgG antibody (goat anti-rabbit IgG conjugated to HRP, Invitrogen, 31462) at 1:5000. The raw blots can be found in Fig. S1.

### Meiotic chromosome spreads from *Mus musculus* testes
This procedure was carried out as described in detail previously (Palmer et al., 2020). The antibodies used were as follows: anti-SYCP3 (Santa Cruz Biotechnology, sc-20845; 1:150), anti-SYCP1 (Abcam, ab15087; 1:150),

anti-YH2AX (EMD Millipore, 05-636; 1:500) and anti-Speedy A (Tu et al., 2017; 1:300).

## Fertility analyses

To assess fertility, animals at 8-10 weeks of age were bred as single pairs. Male $Cdk2^{SHORT/SHORT}$, $Cdk2^{LONG/LONG}$, $Cdk2^{+/SHORT}$ and $Cdk2^{+/LONG}$ mice were paired with the females of the same genotype. The number of pups, gender and genotype in each birth was recorded. Calculations for cumulative born pups were made based on the number of pups born to each breeding pair and plotted as standard deviation in the graph. Percentages of gender and genotype represent the ratio of pups born to a breeding pair for a given trait. To calculate goodness of fit to Mendelian ratio, the $\chi^2$ test was performed based on the expected gender ratios of 50%/50% for female/male and 25%/50%/25% for homozygous mutant/heterozygous/homozygous wild-type mice.

## Image and data analyses

Image and data is described in the figure legends. For Fig. 6S-X, total H2AX fluorescence intensity quantification was performed in Fiji (ImageJ; version 2.16.0/1.54p). For each image, a region of interest encompassing the entire spread was generated using the 'Tracing Wand' tool. Within each region of interest, the mean grey value (representing mean fluorescence intensity of γH2AX) was measured. All images were analysed using constant threshold and brightness settings across groups and stages. Statistical analysis was conducted using BioRender.com, which runs analyses in R (version 4.2.2). For each meiotic stage, normality was assessed using the Shapiro–Wilk test and variance equality with Levene's test. Based on these results, leptotene, zygotene, early pachytene, late pachytene, and diplotene were analysed using the Kruskal–Wallis test, followed by Dunn's post-hoc test; mid-pachytene was analysed using Welch's ANOVA with Games–Howell multiple comparisons. For leptotene (Kruskal–Wallis and Dunn's): wild type versus $Cdk2^{SHORT/SHORT}$, $P$=0.041; $SHORT/SHORT$ versus $LONG/LONG$, $P$=0.0013. For zygotene (Kruskal–Wallis and Dunn's): $SHORT/SHORT$ versus $LONG/LONG$, $P$=0.0018. For early pachytene (Kruskal–Wallis and Dunn's): $SHORT/SHORT$ versus $LONG/LONG$, $P$=0.0018. for mid pachytene (Welch's ANOVA and Games–Howell): wild type versus $SHORT/SHORT$, $P$=0.005, diff=2.67; $SHORT/SHORT$ versus $LONG/LONG$, $P$=0.035, diff=−2.23. For late pachytene (Kruskal–Wallis and Dunn's): wild type versus $SHORT/SHORT$, $P$=0.0195. For diplotene (Kruskal–Wallis and Dunn's), wild type versus $SHORT/SHORT$, $P$=0.0124; $SHORT/SHORT$ versus $LONG/LONG$, $P$=0.0255. Sample sizes ($n$) per group and meiotic stage were: leptotene, $n$=10, 12, 10; zygotene, $n$=5, 8, 8; early-pachytene, $n$=10, 7, 18; mid-pachytene, $n$=10, 21, 6; late-pachytene, $n$=9, 7, 5; diplotene, $n$=13, 13, 14 (wild type, $Cdk2^{SHORT/SHORT}$, $Cdk2^{LONG/LONG}$).

For Fig. 6Y,Z, colocalization of H2AX and SYCP3 was analysed using Fiji (ImageJ; version 2.16.0/1.54p). Each image was converted to 16-bit greyscale, and then the γH2AX image was loaded as channel 1, and the corresponding SYCP3 image as channel 2, into the 'Colocalization Threshold' plugin in Fiji. The plugin computes tM2 (thresholded Manders' coefficient for channel 2), representing the fraction of SYCP3 signal overlapping with γH2AX above the defined threshold. All thresholding and brightness settings were kept constant across groups and stages. Only early and mid-pachytene samples were analysed. Statistics were performed in BioRender.com (R version 4.2.2). Normality was assessed using the Shapiro–Wilk test, and variance homogeneity using Levene's test. For both early and mid-pachytene, a Kruskal–Wallis test was used, followed by Dunn's post-hoc test for pairwise comparisons. For early-pachytene (Kruskal–Wallis and Dunn's), wild type versus $Cdk2^{LONG/LONG}$, $P$=0.018. For mid-pachytene (Kruskal–Wallis and Dunn's) this is $P$=0.767 (ns). Sample sizes ($n$) per group and meiotic stage were: early-pachytene, $n$=9, 3, 16; mid-pachytene, $n$=10, 12, 6 (wild type, $Cdk2^{SHORT/SHORT}$, $Cdk2^{LONG/LONG}$).

## Acknowledgements

We thank all the past and present members of the Kaldis lab for their support and input. We also acknowledge the help of the vets and technicians from the Biological Resource Centre (BRC-A.STAR Singapore) for veterinary assistance. Special thanks go to Tan Qing Hui, Shiela Fransisco Margallo, Pangalingan Christie Chrisma Domingo, and Timothy Teck Chiew Chua for animal expertise.

## Competing interests

The authors declare no competing or financial interests.

## Author contributions

Conceptualization: N.P., P.K.; Data curation: N.P., N.E.K.-Y., H.H., U.K., S.Z.A.T., J.R.O., T.T., C.M.F.G., L.N.Z.; Formal analysis: N.P., N.E.K.-Y., H.H., U.K., L.N.Z.; Funding acquisition: K.L., P.K.; Investigation: N.P., N.E.K.-Y., H.H., U.K., S.Z.A.T., J.R.O., T.T., C.M.F.G., L.N.Z.; Methodology: N.P., N.E.K.-Y., H.H., U.K., S.Z.A.T., J.R.O., T.T., C.M.F.G., L.N.Z.; Project administration: P.K.; Resources: K.L., P.K.; Supervision: E.G., P.K.; Validation: N.P., S.Z.A.T., J.R.O., T.T., C.M.F.G., L.N.Z.; Visualization: L.N.Z.; Writing – original draft: N.P., P.K.; Writing – review & editing: N.P., N.E.K.-Y., H.H., U.K., L.N.Z., E.G., P.K.

## Funding

P.K. is supported by the Swedish Research Council (Vetenskapsrådet; 2021-01331), the Swedish Cancer Society (Cancerfonden; 21-1566Pj and 24-3605Pj), the Crafoord Foundation (Crafoordska Stiftelsen; 20220628), the Faculty of Medicine, Lund University (Lunds Universitet), the Swedish Foundation for Strategic Research (Stiftelsen för Strategisk Forskning; Dnr IRC15-0067), and Swedish Research Council, Strategic Research Area EXODIAB (Dnr 2009-1039). K.L. was supported by a grant from the Shenzhen Science and Technology Innovation Program (KQTD20190929172749226), China. The funders had no role in study design, data collection and interpretation, or the decision to submit this manuscript for publication. Open Access funding provided by Lund University. Deposited in PMC for immediate release.

## Data and resource availability

All relevant data and details of resources can be found within the article and its supplementary information.

## Peer review history

The peer review history is available online at https://journals.biologists.com/jcs/lookup/doi/10.1242/jcs.264291.reviewer-comments.pdf

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
