## [Peer Review File · Journal of Cell Science]

Differential splice isoforms of mouse CDK2 play functionally redundant roles during mitotic and meiotic division

Nathan Palmer, Nisan Ece Kalem-Yapar, Hanna Hultén, Umur Keles, S. Zakiah A. Talib, Jin Rong Ow, Tommaso Tabaglio, Christine Goh, Li Na Zhao, Ernesto Guccione, Kui Liu and Philipp Kaldis

DOI: 10.1242/jcs.264291

Editor: Renata Basto

Review timeline

Submission to Review Commons:	3 September 2024
Submission to Journal of Cell Science:	10 July 2025
Editorial decision:	11 August 2025
First revision received:	4 September 2025
Accepted:	9 September 2025

Reviewer 1

Evidence, reproducibility and clarity

In the manuscript entitled "Differential splice isoforms of mouse CDK2 play functionally redundant roles during mitotic and meiotic division", the authors explore potential differential roles of two splice variants of the Cdk2 kinase, namely Cdk2L and Cdk2S isoforms, expressed in some mouse tissues. To this end, they develop an elegant approach by generating mice constitutively expressing only the shortest Cdk2S (CRISPR KO) or the longest Cdk2L (CRISPR KI) Cdk2 isoform.

Briefly, the authors report that Cdk2L or Cdk2S mice are viable and fully fertile, in contrast to Cdk2^{-/-} mice that are sterile. They further observed no visible alteration of both reproductive organs, testis and ovary. In spermatocytes, both Cdk2 isoforms were correctly located at telomeres and at recombination nodules that promote meiotic crossovers. Supporting that the recombination is effective, they report the transient nature of γ -H2AX foci, a marker of DNA intermediates, along paired chromosomes during meiotic prophase in the different genetic contexts. They showed that the specific meiosis Cdk2 partner Speedy is correctly located at telomeres upon Cdk2L or Cdk2S expression. Finally, they develop a specific antibody against the Cdk2L isoform and report that in WT mice, Cdk2L is mostly located at telomeres, while it can be recruited at recombination nodules and compensate for the absence of Cdk2S expression. Altogether, they conclude that these two Cdk2 isoforms can compensate for each other for mouse viability as well as for fertility.

Major comments:

1. Although IF images provided are of good quality, quantifications are frequently missing. In particular, could the authors provide a quantification of γ -H2AX foci along the stages of meiotic prophase I to reinforce their conclusion that the repair of DNA intermediates does not differ between the different Cdk2 contexts.
2. A Western blot on WT mice tissues showing the specificity of their novel Cdk2L antibody for Cdk2L versus Cdk2S is currently missing. Because Cdk2 is globally more enriched at telomeres than recombination nodules, can the authors exclude that their novel Cdk2L antibody has a lower sensitivity for Cdk2, explaining the apparent preferential location of Cdk2L at telomeres versus nodules in prophase I (Figure 9).

3. A comparison of the immunoprecipitation of Cdk2 partners, notably Cyclin E1/E2, CyclinA2 and Speedy from Cdk2S and Cdk2L mice tissues is currently missing.

Optional

Because in the present work the authors obtained mice animals expressing only Cdk2L versus Cdk2S, immunoprecipitation followed by mass spectrometry analyses of protein partners will certainly be of interest.

Significance

Globally, I found that this manuscript is well written and that experiments displayed were well designed, although the conclusions of some of them were only based on qualitative analyses. This work will be of broad interest in the meiosis and cell cycle fields.

Reviewer 2

Evidence, reproducibility and clarity

Palmer et al. show that mouse CDK2 encodes two isoforms: CDK2S, which is ubiquitous, and CDK2L, primarily found in germ cells. Notably, their research using mouse models indicates that both CDK2L and CDK2S can independently support mitotic and meiotic cell division. This redundancy helps explain why humans can tolerate the evolutionary loss of CDK2L expression.

Significance

Overall, these findings present a compelling narrative regarding CDK2 regulation. The paper offers a clear and concise account of novel results, with experiments laid out in a logical and straightforward manner. While additional experiments or further quantification could be suggested, the authors have established a sound model with significant novelty. This paper is well-suited for publication.

Reviewer 3

Evidence, reproducibility and clarity

The manuscript tests the role of two key isoforms of CDK2 that are present in mice in gametogenesis. Whereas CDK2 is not essential in somatic tissues, it is essential for meiosis during gametogenesis. Alternative splicing generates a short and a long isoforms of CDK2. Both forms are expressed in meiosis in mice but not in humans. It has been a long-standing question if the isoforms have distinct functions. The manuscript addressed this question by generating mice that express only the short version (CDK2S) or long version (CDK2L) of CDK2. Both mutant mice are fully fertile and phenotypic analysis of these mutant mice does not identify meiotic features that are supported by only one of the isoforms. Hence, the authors conclude that both CDK2S and CDK2L are able to perform the essential functions of CDK2 in meiosis.

Major comments

The presented experiments are conclusive and high quality. They support the key conclusion of the manuscript. Nonetheless, there are two major points where the manuscript should be improved.

1. Statistics of comparisons between genotypes should be shown for testes, body weight measurements and assessment of breeding performance and the Mendelian ratios of genotypes (fig. 5). Furthermore, a more detailed description of quantifications of experiments would help the readers to judge the reproducibility of the results (e.g. how many cells or mice were analyzed in each experiment).

Statistical comparisons of testis and bodyweights (the later data is currently not shown, but they should be included) in wild type versus mutants would be particularly important as these

measurements are contradictorily explained in the results and discussion, and they are directly relevant to the main conclusion of the manuscript as detailed in point b.

2. Fig 5a potentially shows a small decrease in testis size in Cdk2L mice as compared to WT and CDK2S. In the results, it is stated that the testis size is the same as in the WT, but in the discussion it is stated that testis of CDK2L is smaller than testes of WT or CDK2S mice. The body weight of the mutant is also described contradictorily in the results and discussion.

It is an important question if smaller testis in CDK2L is due to a defect in meiosis. Is there elevated apoptosis in the CDK2L mutant where the CDK2S form is missing? The CDK2S form seems to be primarily enriched at CO sites as opposed to telomeres according to the manuscript. Is it possible that, in the absence of CDK2S, MLH1 foci numbers are slightly reduced leading to chromosome alignment defects and resultant apoptosis in a subset of metaphase I cells of CDK2L mice? Such defects could result in a slightly reduced testis weight. If a minor CO deficiency existed in the CDK2L mutant, it would suggest that the CDK2S form is preferentially responsible for promoting crossovers as opposed to mediating CDK2 function in telomere attachment to nuclear envelop. Such a result would be highly relevant to the topic of the manuscript, hence CO numbers as detected by MLH1/3 should be quantified in chromosome spreads.

Furthermore, it should be tested (e.g. by cleaved PARP or TUNEL staining in histological sections) if apoptosis is elevated in a particular stage of meiosis possibly in both CDK2S and CDK2L, but at least in CDK2L, where the testis size seems to be significantly smaller than in WT.

I note that the major conclusions that CDK2S and CDK2L are largely redundant and can substitute for each other are well supported even without the additional experiments. Hence, if these experiments are not possible because the mice are no longer available, then the manuscript would still deserve publication. However, in this case, the authors should discuss the limitation and incompleteness of the analysis, and that the current observations are consistent with/support a differential role for CDK2S and CDK2L in promoting crossover formation.

These additional statistics (point 1) and experimental analysis (point 2) can be performed fast and with little effort (perhaps up to 2 months), provided that the mice are still available.

Minor comments

There are also minor issues that can be addressed by textual editing or reworking the figures.

1. Figure 3. It would be better to shift the area that is enlarged (or the preleptotene labelling) so that the preleptotene cells that are pointed out in the enlarged image are not cut into half. In reference to figure 3, it is stated that "In testes of Cdk2S and Cdk2L, normal appearance of metaphase I stage primary spermatocytes, round spermatids, and elongating spermatids indicated the correct progression of spermatocytes through both meiotic divisions (Figure 3)." Whereas this statement is perfectly believable given the high fertility of the mutants, figure 3 does not show metaphase I spermatocytes and elongated spermatids in the mutant. Additional sections showing these stages would be important to present. Alternatively, if this is not possible, it should be stated that some of the data is not shown. The images show sections from immature mice. Is there a reason for not showing sections from adults? Adult sections would allow better presentation of elongated spermatids/sperm in the mutants.

2. Figure 5, it would be important to perform statistics to test if the various genotypes are produced in the expected ratios (e.g. chi-square (χ^2) test for goodness of fit could be used). The legend should specify what the error bars and boxes represent. The number of repeats should be also specified in the figure legend.

3. Please, edit the following statement for accuracy: "In this model meiotic arrest occurs due to the failure of homologous recombination between paired homologs [26]." Reference 26 describe a model, where inter-homolog crossover formation fails and spermatocytes are unable to progress beyond a diplotene-like stage for reasons that are poorly understood. Furthermore, in the late stages, DNA breaks seem to be mostly repaired indicative of successful DNA repair by non-crossover recombination. Hence, it is not clear if a general recombination failure is the proximal reason for the arrest in the mouse model of reference 26.

4. The manuscript very often uses the phrase "normal" to describe the phenotypes of the mutants. I suggest rephrasing as it requires extensive quantitative comparisons between wt and mutants to state that a mutant is normal. If quantifications are not presented (in my opinion, they are not needed in light of the overall phenotypes) it is better to state that easily recognizable differences were not observed between wt and mutants regarding synapsis or other aspects of meiosis. As an example for the overuse of the word "normal", I paste a relevant sections from the text: "We

observed normal co-localization of these proteins during meiotic prophase and the formation of 19 well-separated bivalents in addition to the XY pair indicating normal synapsis (Figure 6). As synapsis was found to be normal in CDK2L and CDK2S mice, we infer that these two isoforms of CDK2 are functionally redundant and can each act independently of the other to maintain the normal meiotic functions of CDK2."

5. Please, edit for accuracy: "During meiotic prophase I, strand invasion leads to DNA intermediates which causes apoptosis and developmental arrest in spermatocytes." It can be confusing to state that strand invasions lead to arrest and apoptosis. Disruption of DSB repair and lack of homolog pairing are thought to cause apoptosis and arrest of spermatocytes.

6. Please, check accuracy in sentence: "As the levels of CDK2 rise around P14 when the first meiotic cells are observed, the levels of CDK2S fall [22]." and "interestingly in meiotic tissues, the expression pattern of Speedy A mirrors that of CDK2 [22]." It is not obvious from the figures of the reference if total CDK2 levels rise as cells progress in prophase. Did the authors mean CDK2L, but even in this case the speedy A and CDK2L patterns seem different? Furthermore, the referred paper reported dropping of CDK2S levels in pachytene, which would correspond to developmental timepoint postnatal day 14, but testis was not analyzed from 14 days old mice, hence this is incorrectly stated.

7. Abbreviations are not always spelled out which hinders understanding for readers outside of the field. Could the authors, please, spell out what PCBP-responsive and P30 or P56 means (I assume P meant to represent days postpartum)?

8. There are sentences that are incomprehensible and difficult to understand due to typos, missing punctuation or due to length and complexity. I provide an incomplete list of examples. I urge the authors to doublecheck the text, in case, I missed some of the affected sentences: Complex sentence that would benefit from splitting: "Instead of an effect of affinity, this data could be a result of CDK2 as being part of a complex at the interstitial sites, is less accessible to the antibodies but this does not seem to be the case for the pan-CDK2 antibodies."

Please, review punctuation:

"For each variation of the crosses performed the average litter size was not significantly different from crosses of wild type and normal Mendelian ratios were observed (Figure 5B)."

and

"Speedy A has been proposed to promote the loading of CDK2 onto telomeres [39] similarly the deletion of CDK2 leads to a loss of Speedy A binding at telomeres [26]."

Typo: "Base on the previous knowledge, we stained chromosome spreads with antibodies against SpeedyA and pan-CDK2 to detect all isoforms of CDK2."

Incomplete sentence: "In contrast, in knock-in models of Cdk2 where the action of Cdk2 at telomeres is not perturbed and homolog pairing occurs as normal [26, 32]."

Significance

The study addresses a long-standing and important question about the biology of CDK2 using state of the art mouse genetics. Whereas the results do not reveal key differential functions for the examined isoforms, the results and conclusions are novel, i.e. it is shown that CDK2S and CDK2L isoforms are able to substitute each other in the essential functions of CDK2 in meiosis. The study have basic scientific importance, and because CDK2 is a key player in both mitotic and meiotic divisions, the study will be interesting for researchers in the cell cycle and gametogenesis fields.

Author response to reviewers' comments

Dear editor,

Thank you for the constructive comments from the reviewers. We are immensely pleased that

all three reviewers recommend publication of our manuscript.

In order to further improve our manuscript and to address the issues that the reviewers have raised, we corrected all the points as detailed below (they are highlighted in yellow). There are a few suggested experiments that we could not perform since the CDK2S/CDK2L mice are not in our colony anymore due to funding issues and due to that I have moved institution. We apologize for this in advance.

Since we have addressed a long-standing and important problem in cell cycle regulation and CDK2 biology, we hope that our manuscript can be published.

Best regards,
Philipp Kaldis

1. Although IF images provided are of good quality, quantifications are frequently missing. In particular, could the authors provide a quantification of γ -H2X foci along the stages of meiotic prophase I to reinforce their conclusion that the repair of DNA intermediates does not differ between the different Cdk2 contexts.

Response: We agree with the reviewer and will/have done the quantification for γ -H2AX, which is shown now in Figure 7.

2. A Western blot on WT mice tissues showing the specificity of their novel Cdk2L antibody for Cdk2L versus Cdk2S is currently missing. Because Cdk2 is globally more enriched at telomeres than recombination nodules, can the authors exclude that their novel Cdk2L antibody has a lower sensitivity for Cdk2, explaining the apparent preferential location of Cdk2L at telomeres versus nodules in prophase I (Figure 9).

Response: We agree that a WB with CDK2L antibodies is important and now is shown in Figure 2, which indicated that our home-made antibody is specific for CDK2L and does not recognize CDK2S. Nevertheless, it is good to remind ourselves that WB and IF are very different experiments and the affinity of antibodies may vary in WB and IF. Importantly though, in Figure 9 we demonstrate that the CDK2L-antibodies do not recognize CDK2S, which indicates that this antibody is specific for CDK2L. In panel V Figure 9, the CDK2L-antibody does not detect anything along the chromosomes because in these mice, only CDK2S is expressed. In the WT, panel I Figure 9, the CDK2L-antibodies detect only CDK2 at the telomeric ends while it cannot detect CDK2 at the interstitial sites. Therefore, our results using IF are consistent with our CDK2L- antibodies to be specific for CDK2L while not being able to detect CDK2S.

Reviewer #1 (Evidence, reproducibility and clarity (Required)):

In the manuscript entitled "Differential splice isoforms of mouse CDK2 play functionally redundant roles during mitotic and meiotic division", the authors explore potential differential roles of two splice variants of the Cdk2 kinase, namely Cdk2L and Cdk2S isoforms, expressed in some mouse tissues. To this end, they develop an elegant approach by generating mice constitutively expressing only the shortest Cdk2S (CRISPR KO) or the longest Cdk2L (CRISPR KI) Cdk2 isoform.

Briefly, the authors report that Cdk2L or Cdk2S mice are viable and fully fertile, in contrast to Cdk2^{-/-} mice that are sterile. They further observed no visible alteration of both reproductive organs, testis and ovary. In spermatocytes, both Cdk2 isoforms were correctly located at telomeres and at recombination nodules that promote meiotic crossovers. Supporting that the recombination is effective, they report the transient nature of γ -H2AX foci, a marker of DNA intermediates, along paired chromosomes during meiotic prophase in the different genetic contexts. They showed that the specific meiosis Cdk2 partner Speedy is correctly located at telomeres upon Cdk2L or Cdk2S expression. Finally, they develop a specific antibody against the Cdk2L isoform and report that in WT mice, Cdk2L is mostly located at telomeres, while it can be recruited at recombination nodules and compensate for the absence of Cdk2S expression. Altogether, they conclude that these two Cdk2 isoforms can compensate for each other for mouse viability as well as for fertility.

Response: We thank this reviewer for the positive assessment and support of our work.

Major comments:

3. A comparison of the immunoprecipitation of Cdk2 partners, notably Cyclin E1/E2, CyclinA2 and Speedy from Cdk2S and Cdk2L mice tissues is currently missing. Optional
Because in the present work the authors obtained mice animals expressing only Cdk2L versus Cdk2S, immunoprecipitation followed by mass spectrometry analyses of protein partners will certainly be of interest.

Response: Both comments are excellent and important. We had previously shown that SpeedyA can interact with CDK2L and also with CDK2S (PNAS114_592, supplementary data) and other groups also showed that the affinity for cyclin binding is similar between CDK2L and CDK2S (DNA and Cell Biology20_413).

Identifying the binding partners of CDK2L and CDK2S is brilliant and definitely on our to-do-list but is also clearly beyond the scope of this manuscript since it is a new project that could not be completed within the timeframe of the revisions.

Reviewer #1 (Significance (Required)):

Globally, I found that this manuscript is well written and that experiments displayed were well designed, although the conclusions of some of them were only based on qualitative analyses. This work will be of broad interest in the meiosis and cell cycle fields.

Response: We thank the reviewer for the support of our manuscript.

Reviewer #2 (Evidence, reproducibility and clarity (Required)):

Palmer et al. show that mouse CDK2 encodes two isoforms: CDK2S, which is ubiquitous, and CDK2L, primarily found in germ cells. Notably, their research using mouse models indicates that both CDK2L and CDK2S can independently support mitotic and meiotic cell division. This redundancy helps explain why humans can tolerate the evolutionary loss of CDK2L expression.

Reviewer #2 (Significance (Required)):

Overall, these findings present a compelling narrative regarding CDK2 regulation. The paper offers a clear and concise account of novel results, with experiments laid out in a logical and straightforward manner. While additional experiments or further quantification could be suggested, the authors have established a sound model with significant novelty. This paper is well-suited for publication.

Response: We thank this reviewer for the positive assessment and support of our work.

Reviewer #3 (Evidence, reproducibility and clarity (Required)):

1. Evidence, reproducibility, and clarity

The manuscript tests the role of two key isoforms of CDK2 that are present in mice in gametogenesis. Whereas CDK2 is not essential in somatic tissues, it is essential for meiosis during gametogenesis. Alternative splicing generates a short and a long isoforms of CDK2. Both forms are expressed in meiosis in mice but not in humans. It has been a long-standing question if the isoforms have distinct functions. The manuscript addressed this question by generating mice that express only the short version (CDK2S) or long version (CDK2L) of CDK2. Both mutant mice are fully fertile and phenotypic analysis of these mutant mice does not identify meiotic features that are supported by only one of the isoforms. Hence, the authors conclude that both CDK2S and CDK2L are able to perform the essential functions of CDK2 in meiosis.

Major comments

The presented experiments are conclusive and high quality. They support the key conclusion of the manuscript. Nonetheless, there are two major points where the manuscript should be improved.

1. Statistics of comparisons between genotypes should be shown for testes, body weight measurements and assessment of breeding performance and the Mendelian ratios of genotypes (fig. 5). Furthermore, a more detailed description of quantifications of experiments would help the readers to judge the reproducibility of the results (e.g. how many cells or mice were analyzed in each experiment).

Statistical comparisons of testis and bodyweights (the later data is currently not shown, but they should be included) in wild type versus mutants would be particularly important as these

measurements are contradictorily explained in the results and discussion, and they are directly relevant to the main conclusion of the manuscript as detailed in point b.

Response: We thank this reviewer for reminding us to add the statistics in our results in Figure 5. We have done so now and also described it in the methods.

2. Fig 5a potentially shows a small decrease in testis size in Cdk2L mice as compared to WT and CDK2S. In the results, it is stated that the testis size is the same as in the WT, but in the discussion it is stated that testis of CDK2L is smaller than testes of WT or CDK2S mice. The body weight of the mutant is also described contradictorily in the results and discussion.

It is an important question if smaller testis in CDK2L is due to a defect in meiosis. Is there elevated apoptosis in the CDK2L mutant where the CDK2S form is missing? The CDK2S form seems to be primarily enriched at CO sites as opposed to telomeres according to the manuscript. Is it possible that, in the absence of CDK2S, MLH1 foci numbers are slightly reduced leading to chromosome alignment defects and resultant apoptosis in a subset of metaphase I cells of CDK2L mice? Such defects could result in a slightly reduced testis weight. If a minor CO deficiency existed in the CDK2L mutant, it would suggest that the CDK2S form is preferentially responsible for promoting crossovers as opposed to mediating CDK2 function in telomere attachment to nuclear envelop. Such a result would be highly relevant to the topic of the manuscript, hence CO numbers as detected by MLH1/3 should be quantified in chromosome spreads. Furthermore, it should be tested (e.g. by cleaved PARP or TUNEL staining in histological sections) if apoptosis is elevated in a particular stage of meiosis possibly in both CDK2S and CDK2L, but at least in CDK2L, where the testis size seems to be significantly smaller than in WT.

I note that the major conclusions that CDK2S and CDK2L are largely redundant and can substitute for each other are well supported even without the additional experiments.

Hence, if these experiments are not possible because the mice are no longer available, then the manuscript would still deserve publication. However, in this case, the authors should discuss the limitation and incompleteness of the analysis, and that the current observations are consistent with/support a differential role for CDK2S and CDK2L in promoting crossover formation.

These additional statistics (point 1) and experimental analysis (point 2) can be performed fast and with little effort (perhaps up to 2 months), provided that the mice are still available.

Response: We corrected this discrepancy in the description and agree that apoptosis is an important aspect. In addition, quantification of the MLH1/3 foci would be very interesting. Unfortunately, we do not have these mice anymore in our colony at the moment due to funding limitations and it would take us several months to revive the animals to perform these experiments. Furthermore, we consider these experiments to be not absolutely essential since we have shown that both CDK2S and CDK2L animals are fertile. We hope the reviewer will be understanding our difficult situation. To acknowledge this reviewer's point, we have added a few sentences at the end of the discussion (see bottom of page 7).

Minor comments

There are also minor issues that can be addressed by textual editing or reworking the figures.

1. Figure 3. It would be better to shift the area that is enlarged (or the preleptotene labelling) so that the preleptotene cells that are pointed out in the enlarged image are not cut into half. In reference to figure 3, it is stated that "In testes of Cdk2S and Cdk2L, normal appearance of metaphase I stage primary spermatocytes, round spermatids, and elongating spermatids indicated the correct progression of spermatocytes through both meiotic divisions (Figure 3)." Whereas this statement is perfectly believable given the high fertility of the mutants, figure 3 does not show metaphase I spermatocytes and elongated spermatids in the mutant. Additional sections showing these stages would be important to present. Alternatively, if this is not possible, it should be stated that some of the data is not shown. The images show sections from immature mice. Is there a reason for not showing sections from adults? Adult sections would allow better presentation of elongated spermatids/sperm in the mutants.

Response: We have corrected the text and added "Figure 3 and data not shown". We agree with the reviewer that it would be ideal to show adult testis but currently these mice are not alive and I have moved to another institution, which would lead to a long delay of such an experiment.

2. Figure 5, it would be important to perform statistics to test if the various genotypes are produced in the expected ratios (e.g. chi-square (χ^2) test for goodness of fit could be used). The legend should specify what the error bars and boxes represent. The number of repeats should be

also specified in the figure legend.

Response: We agree and have done this (see legend for Figure 5 and also Methods).

3. Please, edit the following statement for accuracy: "In this model meiotic arrest occurs due to the failure of homologous recombination between paired homologs [26]." Reference 26 describe a model, where inter-homolog crossover formation fails and spermatocytes are unable to progress beyond a diplotene-like stage for reasons that are poorly understood. Furthermore, in the late stages, DNA breaks seem to be mostly repaired indicative of successful DNA repair by non-crossover recombination. Hence, it is not clear if a general recombination failure is the proximal reason for the arrest in the mouse model of reference 26.

Response: Thank you for pointing this out to us. We have rephrased those sentences and have avoided overinterpreting the data from reference 26: "In contrast, in knock- in models of Cdk2 where the action of Cdk2 at telomeres is not perturbed and homolog pairing occurs successfully, telomere fusion events increase, leading to apoptosis for several reasons [25, 31]."

4. The manuscript very often uses the phrase "normal" to describe the phenotypes of the mutants. I suggest rephrasing as it requires extensive quantitative comparisons between wt and mutants to state that a mutant is normal. If quantifications are not presented (in my opinion, they are not needed in light of the overall phenotypes) it is better to state that easily recognizable differences were not observed between wt and mutants regarding synapsis or other aspects of meiosis. As an example for the overuse of the word "normal", I paste a relevant sections from the text: "We observed normal co-localization of these proteins during meiotic prophase and the formation of 19 well-separated bivalents in addition to the XY pair indicating normal synapsis (Figure 6). As synapsis was found to be normal in CDK2L and CDK2S mice, we infer that these two isoforms of CDK2 are functionally redundant and can each act independently of the other to maintain the normal meiotic functions of CDK2."

Response: We agree that the usage of 'normal' in mutant mice is unfortunate and we have tried to avoid or reduce this expression throughout the manuscript. For example, "The co-localization of SYCP3 and SYCP1 during meiotic prophase and the formation of 19 well-separated bivalents in addition to the XY pair indicating successful synapsis in both CDK2S and CDK2L testes (Figure 6). As synapsis was comparable in CDK2L and CDK2S mice, we infer that these two isoforms of CDK2 are functionally redundant and can each act independently of the other to maintain the meiotic functions of CDK2."

5. Please, edit for accuracy: "During meiotic prophase I, strand invasion leads to DNA intermediates which causes apoptosis and developmental arrest in spermatocytes." It can be confusing to state that strand invasions lead to arrest and apoptosis. Disruption of DSB repair and lack of homolog pairing are thought to cause apoptosis and arrest of spermatocytes.

Response: We have corrected this "During meiotic prophase I, disruption of double- strand break (DSB) repair and failure of homolog pairing are believed to trigger apoptosis and arrest in spermatocytes."

6. Please, check accuracy in sentence: "As the levels of CDK2 rise around P14 when the first meiotic cells are observed, the levels of CDK2S fall [22]." and "interestingly in meiotic tissues, the expression pattern of Speedy A mirrors that of CDK2 [22]." It is not obvious from the figures of the reference if total CDK2 levels rise as cells progress in prophase. Did the authors mean CDK2L, but even in this case the speedy A and CDK2L patterns seem different? Furthermore, the referred paper reported dropping of CDK2S levels in pachytene, which would correspond to developmental timepoint postnatal day 14, but testis was not analyzed from 14 days old mice, hence this is incorrectly stated.

Response: Thank you - we have corrected this accordingly "The relative expression CDK2S and CDK2L in testis also changes during development. CDK2S is detectable starting from P8 until the end of pachytene and is almost undetectable in round and elongated spermatids (see Figure 1B-C in [21]). In contrast, CDK2L appears at P12 and is expressed from then on the same than CDK2S. Interestingly, Speedy A starts to be expressed at the same time than CDK2L at P12 (see Figure 1C in [21]). The localization of CDK2L to telomeres seem to be dependent on binding to Speedy A (see Figure S8C-L in [21]) and probably on CDK2L activity."

7. Abbreviations are not always spelled out which hinders understanding for readers outside of the field. Could the authors, please, spell out what PCBP-responsive and P30 or P56 means (I

assume P meant to represent days postpartum)?

Response: We have now spelled out all the abbreviations. In addition, we removed the part about "PCBP-responsive" because it was distracting.

8. There are sentences that are incomprehensible and difficult to understand due to typos, missing punctuation or due to length and complexity. I provide an incomplete list of examples. I urge the authors to doublecheck the text, in case, I missed some of the affected sentences: Complex sentence that would benefit from splitting: "Instead of an effect of affinity, this data could be a result of CDK2 as being part of a complex at the interstitial sites, is less accessible to the antibodies but this does not seem to be the case for the pan-CDK2 antibodies."

Response: We agree and have corrected this accordingly "Although it became quickly clear that CDK2S and CDK2L mice were fertile, we investigated several stages of meiosis and all the known functions of CDK2 but ultimately had to conclude that CDK2S and CDK2L can compensate for each other's functions at least in the laboratory setting. One of the potential interesting information was that the specific antibodies against CDK2L we had generated, stained the telomeres very well, as expected, but the staining of the interstitial sites was low to undetectable. This confirmed that CDK2L has a high affinity for the telomeres but there was still CDK2 localized to the interstitial sites in the absence of CDK2S (Figure 9, IX-XII) detected by the pan-CDK2 antibodies. The importance of these findings lays in the specificity of our antibodies against CDK2L (see also Methods), which detect CDK2L but not CDK2S (Figure 9, V-VIII). We do not believe this is an effect of affinity of the CDK2L antibodies, but we cannot exclude the remote possibility that CDK2 is part of a complex at the interstitial sites, which block accessibility of the CDK2L antibodies but not of the pan-CDK2 antibodies. Therefore, we conclude that CDK2L is mostly localized to telomeres but retains low affinity to the interstitial sites. Of course, the functional consequences of the localization of CDK2L to telomeres needs to be further investigated in the future."

Please, review punctuation:

"For each variation of the crosses performed the average litter size was not significantly different from crosses of wild type and normal Mendelian ratios were observed (Figure 5B)." and "Speedy A has been proposed to promote the loading of CDK2 onto telomeres [39] similarly the deletion of CDK2 leads to a loss of Speedy A binding at telomeres [26]."

Response: This has been corrected.

Typo: "Base on the previous knowledge, we stained chromosome spreads with antibodies against SpeedyA and pan-CDK2 to detect all isoforms of CDK2."

Response: This has been corrected.

Incomplete sentence: "In contrast, in knock-in models of Cdk2 where the action of Cdk2 at telomeres is not perturbed and homolog pairing occurs as normal [26, 32]."

Response: This has been corrected.

Reviewer #3 (Significance (Required)):

2. Significance

The study addresses a long-standing and important question about the biology of CDK2 using state of the art mouse genetics. Whereas the results do not reveal key differential functions for the examined isoforms, the results and conclusions are novel, i.e. it is shown that CDK2S and CDK2L isoforms are able to substitute each other in the essential functions of CDK2 in meiosis. The study have basic scientific importance, and because CDK2 is a key player in both mitotic and meiotic divisions, the study will be interesting for researchers in the cell cycle and gametogenesis fields.

Response: We thank the reviewer for the support of our work.

Original submission

First decision letter

MS ID#: jcs.264291

MS TITLE: Differential splice isoforms of mouse CDK2 play functionally redundant roles during mitotic and meiotic division

AUTHORS: Nathan Palmer; Nisan Ece Kalem-Yapar; Hanna Hultén; Umur Keles; Zakiah A. Talib; Jin Rong Ow; Tommaso Tabaglio; Christine Goh; Li Na Zhao; Ernesto Guccione; Kui Liu; Philipp Kaldis
ARTICLE TYPE: Review Commons Transfer

Dear Dr Kaldis,

We have now reached a decision on the above manuscript.

As you will see, the reviewers gave favourable reports but raised a few points that will require amendments to your manuscript. I hope that you will be able to carry these out because I would like to be able to accept your paper.

Thank you for sending your manuscript to Journal of Cell Science through Review Commons.

Reviewer 1: SUMMARY OF THE ADVANCE MADE IN THIS PAPER AND ITS POTENTIAL SIGNIFICANCE TO THE FIELD

Manuscript Number: jcs.264291

Title: Differential splice isoforms of mouse CDK2 play functionally redundant roles during mitotic and meiotic division

This manuscript by Palmer et al. investigates a long-standing question regarding the *in vivo* functions of two naturally occurring splice isoforms of Cyclin-Dependent Kinase 2 (CDK2) in mice: the shorter, constitutively expressed CDK2S and the longer CDK2L, which is preferentially expressed during meiosis. While CDK2 is known to be dispensable for mitotic division but essential for meiosis, the specific roles of these isoforms have remained unclear. Using an elegant genetic approach with CRISPR/Cas9, the authors generated two novel mouse models: one that exclusively expresses CDK2S (Cdk2SHORT/SHORT) and another that exclusively expresses CDK2L (Cdk2LONG/LONG). The central finding is that both mouse lines are viable and fully fertile. Through detailed histological and cytological analysis of spermatogenesis, the authors demonstrate that either isoform alone is sufficient to support all essential meiotic processes, including synapsis, DNA double-strand break repair, and the localization of key binding partners like SpeedyA. While the authors identify a preferential localization of CDK2L to telomeres in wild-type mice, they show that each isoform can functionally compensate for the other's absence at all required chromosomal locations.

SUGGESTIONS TO AUTHORS

The authors have done an excellent job addressing the previous major concerns. The addition of quantifications for γ H2AX foci (Figure 7), the validation of the CDK2L-specific antibody (Figure 2), and the thoughtful discussion of the study's limitations have significantly improved the manuscript. The central conclusion—that CDK2S and CDK2L are functionally redundant—is now very well-supported.

While the authors have addressed the discrepancy regarding testis size in CDK2L mice, the observation that these mice are slightly smaller remains intriguing. The authors correctly note that exploring the molecular basis for this mild phenotype would be challenging and is beyond the scope of the current study. Their decision to acknowledge this and other potential avenues for future research (e.g., analyzing apoptosis, quantifying MLH1/3 foci) in the discussion is appropriate and sufficient. It is disappointing to read that the CDK2S and CDK2L strains are not in their colony anymore, as this would have been a very interesting avenue to pursue and close the investigation. Nevertheless, it does not detract from the excellent work presented in this paper.

Minor comments:

In the abstract it is stated, "...conventional knockout methodology leads to the loss of both isoforms." and then, "In this study, by generating mice that only express CDK2S or CDK2L...". This is slightly redundant. It could be streamlined to something like: "As conventional knockout methodology eliminates both isoforms, we generated mice expressing only CDK2S or CDK2L to dissect their individual roles."

The abstract and introduction note that the two isoforms exhibit different kinase activities in vitro. The discussion could be slightly enhanced by briefly revisiting this point. A sentence could be added to speculate why these in vitro differences in kinase activity do not translate into an observable fertility phenotype in vivo. For example, the authors might suggest that in the cellular context, factors such as protein concentration, subcellular localization, and substrate accessibility are more critical determinants of function than the subtle differences in intrinsic enzymatic activity measured in vitro. This is not an essential revision, but could add a nice concluding thought.

Reviewer 2: SUMMARY OF THE ADVANCE MADE IN THIS PAPER AND ITS POTENTIAL SIGNIFICANCE TO THE FIELD

This is a revised manuscript from Philipp Kaldis' group describing the role of two CDK2 splice isoforms in mammalian meiosis. The data convincingly support the main conclusions, and the authors have adequately addressed the previous concerns raised by the reviewers. This is an important contribution to the field, and I support the publication of this manuscript.

SUGGESTIONS TO AUTHORS

I only have minor comments, which can be addressed through textural revisions.

Page 2, line 21, please correct the typo "fulfil" to "fulfill".

Page 2, line 35, references need to be combined.

Page 2, line 37, change "8" to "eight" for consistency.

Page 3, line 46, The phrase "Cdk2^{-/-} (lacking both CDK2S and CDK2L) heterozygote mice" is confusing as "-/-" denotes homozygosity.

Page 4, line 49, the subheading "Repair of DNA intermediates" should be revised to "Repair of recombination intermediates", as the term "DNA intermediates" is inaccurate and unclear what it refers to.

Page 5, line 47, please change "at the same time than" to "at the same time as".

First revisionAuthor response to reviewers' comments

Dear editor,

We are grateful for the constructive comments from the reviewers. We are immensely pleased that both reviewers recommend publication of our manuscript.

To further improve our manuscript further and to address the minor issues that the reviewers

have raised, we corrected them all as detailed below.

Best regards,

Philipp Kaldis

Reviewer 1: SUMMARY OF THE ADVANCE MADE IN THIS PAPER AND ITS POTENTIAL SIGNIFICANCE TO THE FIELD

Title: Differential splice isoforms of mouse CDK2 play functionally redundant roles during mitotic and meiotic division

This manuscript by Palmer et al. investigates a long-standing question regarding the *in vivo* functions of two naturally occurring splice isoforms of Cyclin-Dependent Kinase 2 (CDK2) in mice: the shorter, constitutively expressed CDK2S and the longer CDK2L, which is preferentially expressed during meiosis. While CDK2 is known to be dispensable for mitotic division but essential for meiosis, the specific roles of these isoforms have remained unclear. Using an elegant genetic approach with CRISPR/Cas9, the authors generated two novel mouse models: one that exclusively expresses CDK2S (Cdk2SHORT/SHORT) and another that exclusively expresses CDK2L (Cdk2LONG/LONG). The central finding is that both mouse lines are viable and fully fertile. Through detailed histological and cytological analysis of spermatogenesis, the authors demonstrate that either isoform alone is sufficient to support all essential meiotic processes, including synapsis, DNA double-strand break repair, and the localization of key binding partners like SpeedyA. While the authors identify a preferential localization of CDK2L to telomeres in wild-type mice, they show that each isoform can functionally compensate for the other's absence at all required chromosomal locations.

The authors have done an excellent job addressing the previous major concerns.

The addition of quantifications for γ H2AX foci (Figure 7), the validation of the CDK2L-specific antibody (Figure 2), and the thoughtful discussion of the study's limitations have significantly improved the manuscript. The central conclusion—that CDK2S and CDK2L are functionally redundant—is now very well-supported.

While the authors have addressed the discrepancy regarding testis size in CDK2L mice, the observation that these mice are slightly smaller remains intriguing. The authors correctly note that exploring the molecular basis for this mild phenotype would be challenging and is beyond the scope of the current study. Their decision to acknowledge this and other potential avenues for future research (e.g., analyzing apoptosis, quantifying MLH1/3 foci) in the discussion is appropriate and sufficient. It is disappointing to read that the CDK2S and CDK2L strains are not in their colony anymore, as this would have been a very interesting avenue to pursue and close the investigation. Nevertheless, it does not detract from the excellent work presented in this paper.

We thank this reviewer for the positive assessment and support of our work.

Minor comments:

In the abstract it is stated, "...conventional knockout methodology leads to the loss of both isoforms." and then, "In this study, by generating mice that only express CDK2S or CDK2L...". This is slightly redundant. It could be streamlined to something like: "As conventional knockout methodology eliminates both isoforms, we generated mice expressing only CDK2S or CDK2L to dissect their individual roles."

We agree and change it within the limits of abstract word count. "...as conventional knockout methodology deletes both of the isoforms, we generated mice expressing only CDK2S or CDK2L and found that both CDK2L and CDK2S are sufficient to support both mitotic and meiotic division when expressed in the absence of the other.

The abstract and introduction note that the two isoforms exhibit different kinase activities *in vitro*. The discussion could be slightly enhanced by briefly revisiting this point. A sentence could be added to speculate why these *in vitro* differences in kinase activity do not translate into an observable fertility phenotype *in vivo*. For example, the authors might suggest that in the cellular context, factors such as protein concentration, subcellular localization, and substrate

accessibility are more critical determinants of function than the subtle differences in intrinsic enzymatic activity measured in vitro. This is not an essential revision, but could add a nice concluding thought.

We agree that it would add a nice concluding remark and added: “This resulted in two mouse stains, CDK2S and CDK2L, that are viable and fully fertile, despite that CDK2L and CDK2S display differences in kinase activity in vitro. This may reflect CDK2's essentiality during the meiotic cell cycle and therefore may imply that the gene function does not exclusively depend on enzyme activity, but other factors such as substrate accessibility and subcellular localization, however, more studies will need to be performed in that regard.”

Reviewer 2: SUMMARY OF THE ADVANCE MADE IN THIS PAPER AND ITS POTENTIAL SIGNIFICANCE TO THE FIELD

This is a revised manuscript from Philipp Kaldis' group describing the role of two CDK2 splice isoforms in mammalian meiosis. The data convincingly support the main conclusions, and the authors have adequately addressed the previous concerns raised by the reviewers. This is an important contribution to the field, and I support the publication of this manuscript.

We thank this reviewer for the positive assessment and support of our work.

SUGGESTIONS TO AUTHORS

I only have minor comments, which can be addressed through textual revisions.

Page 2, line 21, please correct the typo "fulfil" to "fulfill".

The typo has been corrected. ✓

Page 2, line 35, references need to be combined.

References are combined. ✓

Page 2, line 37, change "8" to "eight" for consistency.

This has been corrected. ✓

Page 3, line 46, The phrase "Cdk2^{-/-} (lacking both CDK2S and CDK2L) heterozygote mice" is confusing as "-/-" denotes homozygosity.

This has been corrected. ✓

Page 4, line 49, the subheading "Repair of DNA intermediates" should be revised to "Repair of recombination intermediates", as the term "DNA intermediates" is inaccurate and unclear what it refers to.

This has been corrected. ✓

Page 5, line 47, please change "at the same time than" to "at the same time as".

This has been corrected. ✓

Second decision letter

MS ID#: jcs.264291R1

MS Title: Differential splice isoforms of mouse CDK2 play functionally redundant roles during mitotic and meiotic division

Authors: Nathan Palmer; Nisan Ece Kalem-Yapar; Hanna Hultén; Umur Keles; Zakiah A. Talib; Jin Rong Ow; Tommaso Tabaglio; Christine Goh; Li Na Zhao; Ernesto Guccione; Kui Liu; Philipp Kaldis
Article Type: Review Commons Transfer

Dear Dr Kaldis,

I am happy to tell you that your manuscript has been accepted for publication in Journal of Cell Science, pending standard publication integrity checks.